# Biomimetic Systems Involving Macrophages and Their Potential for Targeted Drug Delivery

**DOI:** 10.3390/biomimetics8070543

**Published:** 2023-11-12

**Authors:** Ivan V. Savchenko, Igor D. Zlotnikov, Elena V. Kudryashova

**Affiliations:** Faculty of Chemistry, Lomonosov Moscow State University, Leninskie Gory, 1/3, 119991 Moscow, Russiazlotnikovid@my.msu.ru (I.D.Z.)

**Keywords:** macrophage-mediated therapy, macrophage biomimetics, macrophage-derived particles, selective ligands

## Abstract

The concept of targeted drug delivery can be described in terms of the drug systems’ ability to mimic the biological objects’ property to localize to target cells or tissues. For example, drug delivery systems based on red blood cells or mimicking some of their useful features, such as long circulation in stealth mode, have been known for decades. On the contrary, therapeutic strategies based on macrophages have gained very limited attention until recently. Here, we review two biomimetic strategies associated with macrophages that can be used to develop new therapeutic modalities: first, the mimicry of certain types of macrophages (i.e., the use of macrophages, including tumor-associated or macrophage-derived particles as a carrier for the targeted delivery of therapeutic agents); second, the mimicry of ligands, naturally absorbed by macrophages (i.e., the use of therapeutic agents specifically targeted at macrophages). We discuss the potential applications of biomimetic systems involving macrophages for new advancements in the treatment of infections, inflammatory diseases, and cancer.

## 1. Introduction

Due to the increased risk of infectious diseases, with the lack of ways to effectively treat oncology, the need to develop and improve treatment methods, in particular, drug therapy, increases. However, drug therapy of inflammatory diseases, including oncology and infectious diseases, has limitations, since drugs in current use often have significant drawbacks, such as toxicity to healthy tissues, immunoreactivity, a short circulation time and low stability in the biological media. In this regard, systems of targeted drug delivery to the pathology area (site of inflammation or tumor) and/or to individual pathogens (viruses, bacteria, parasites, etc.) are of the greatest interest. Due to the selectivity of the drug in this case, it is possible to avoid its effect on healthy tissues and organs of the body and reduce the effective dose required for treatment.

Thanks to the use of nanoparticles, it became possible to partially overcome such problems as the low solubility of the drug in biological fluids, the low stability of biodegradable therapeutic agents and their toxic effect on biological systems [1,2,3]. However, despite the high ability of nanoparticles to cross many biological barriers and diffuse in intercellular and cellular media, the development of a targeted drug delivery method is still required. Recently, cell-mediated drug delivery using red blood cells, neutrophils, macrophages, stem cells and lymphocytes has attracted much attention, due to its multifunctionality and inherent stability in biological media.

During the course of diseases and inflammatory processes, a complex of immune reactions takes place in the organizm, where infiltration and the targeted transportation of immune system cells—leukocytes—are primary. In particular, macrophages play an active role in the course of the immune response, the main function of which is the selective engulfment and removal of pathogens and necrosis products of cells, as well as the activation of lymphocytes in the site of inflammation [4,5]. Since macrophages play an important role in a variety of pathological processes, many biomimetic approaches inspired by the biological functions of macrophages have recently appeared. Among them, two distinct macrophage-mediated therapeutic strategies have been developed: the application of macrophage-derived particles for targeted therapy, and the use of therapeutic agents specifically targeting macrophages in vivo (Figure 1).

The first strategy is based on the targeting ability of macrophages, which is due to their functional proteins, namely Pattern-Recognition Receptors (PRRs), such as TLR4 and CD14, receptors, which bind proinflammatory cytokines, such as CD120, CD126 and CD119, and receptors that bind factors of the inflamed endothelium, such as CD44 and Mac-1. Applications of therapeutic agents, such as drugs, nucleic acids, and proteins encapsulated into macrophage-derived particles or adhered to macrophages has proved to be a potent strategy to treat various diseases, especially cancer and infections, due to the targeting ability, affinity to cytokines and biocompatibility of these biomimetic formulations inherited from macrophages. In this regard, biomimetically designed drug delivery systems using macrophages or macrophage-derived particles exhibit a prolonged drug circulation time, improved biocompatibility, decreased immunogenicity and enhanced targeting ability inherited from macrophages. 

The potency of the second strategy is based on the macrophages’ relation with the development of pathological processes. The plasticity of a macrophage has crucial role in the occurrence and development of various chronic diseases, such as atherosclerosis and cancer. Moreover, some diseases, such as HIV infection, tuberculosis, leishmaniasis, granulomatosis, atypical pneumonia caused by *Chlamydia pneumoniae*, etc., are caused by microorganisms that use macrophages as safe reservoirs, which reduces the exposure to chemotherapy and prevents immune detection. Therefore, the macrophage has become an important therapeutic target. In this regard, immunomodulators, such as cytokines, siRNA, macrophage receptor agonists/antagonists and some other therapeutic agents have attracted attention. In addition, with the development of nanotechnologies, drug-loaded nanoparticles are presented to be potential macrophage-targeted therapeutic agents, especially when modified with ligands, such as mannose, galactose, glucans, hyaluronic acid and others, for enhanced macrophage uptake via recognition by macrophage receptors.

## 2. Macrophages as Immune System Cells

Macrophages are present in almost all tissues of the body [6]. The origin of macrophages in different tissues is not fully established [7,8,9], although it is generally believed that some macrophages originate during embryonic development, and other macrophages represent a mature form of monocytes [10,11,12,13]. The lifetime of macrophages ranges from a few days to several years, and during this time they perform several different functions within the framework of innate and adaptive immunity [6,14]. 

The phagocytic function of macrophages consists of the absorption and utilization of pathogens and infected cells and can be carried out both without the participation of other cells of the immune system, and within the framework of the complement system; in both cases, macrophages act through specific receptors. Macrophages also play an important role in the participation and regulation of immune responses—macrophages induce inflammation by producing a variety of inflammatory mediators that activate cells of the immune system and involve them in the immune response.

Macrophages represent an important class of sensory cells capable of detecting pathogens and initiating an immune response through mediators. This is possible due to the presence of pattern recognition receptors (PRRs), which detect small molecules and/or regular pathogen-associated molecular patterns (PAMPs). These structures are usually mannose-rich oligosaccharides, peptidoglycan and lipopolysaccharides of the bacterial cell wall, as well as nucleic acids. Among PRRs, membrane-bound Toll-like receptors (TLRs) and scavenger receptors (SRs) play a significant role in phagocytosis and regulating immune responses. 

### 2.1. Two Macrophage Phenotypes

Due to the variety of functions performed, two major phenotypes are distinguished among macrophages: M1- and M2-macrophages [15,16]. It is known that macrophages are highly plastic cells and can change the phenotype depending on various environmental factors, such as cytokines, pathogens and stress factors [16,17,18,19]. Information about well-defined macrophage phenotypes is summarized in Table 1.

M1 macrophages are the most characterized subpopulation and are known primarily for their phagocytic function [20,21]. These macrophages are activated during cell-mediated immune reactions through the action of stress factors [22,23] and cytokines, mainly IFN-γ [24], TNF-α [25], and the recognition of PAMPs via TLRs [26,27]. Activation provokes the secretion of various cytokines (TNF-α, IFN-γ, IL-1, IL-6, IL-12, and IL-23) [28,29,30] and enzymes (MMPS, MMP12, hyaluronidase, and collagenase) [31,32,33,34] by macrophages, which leads to an amplification of the immune response and destruction of the extracellular matrix. The main functions of M1 macrophages are phagocytosis, the utilization of the remnants of destroyed cells, regulation of the immune response, presentation of antigen and destruction of the extracellular matrix for tissue reorganization.

M2 macrophages are a subpopulation of alternatively activated macrophages [35], namely by IL-13 [36,37], IL-10 [38], TGF-β [39], glucocorticoids [40] and some other factors [41,42]. A distinctive feature of M2 macrophages is their homeostatic, regenerative and anti-inflammatory functions, which is due to their production of STAB-1 [43], IL-10 [44,45], fibronectin, collagen [46,47] and IL-1 receptor antagonists [48]. In addition to the defined M2 phenotype, it has subpopulations: M2a, activated by IL-4; M2b activated by immune complexes LPS and IL-1β; and M2c, activated by IL-10 and TGF-β1. M2a and M2c macrophages are responsible mainly for regenerative processes, such as matrix remodeling and angiogenesis, while M2b macrophages participate in immune regulation and T helper 2 activation.

**Table 1 biomimetics-08-00543-t001:** Macrophage phenotypes of different polarization statuses: pro-inflammatory and anti-inflammatory.

	M1	M2a	M2b	M2c
Activation factors	IFN-γ, LPS, TNF-α	IL-4, IL-13	Immune complexes, LPS, IL-1β	IL-10, TGF-β1
Surface markers	CD80, CD86, TLR2, TLR4, MHC-II	CD163, CD206, MHC-II	CD86, MHC-II	CCR2, CD163, TLR1, TLR8
Secreted cytokines	IL-1, IL-6, IL-12, IL-23, TNF-α	IL-10, IL-1Ra, TGF-β	IL-1, IL-6, IL-10, TNF-α	IL-10, TGF-β
Functions	Inflammatory response, phagocytosis of debris/cells, antigen presentation, stimulation of vascular sprouting	Inflammatory response, matrix deposition, fibrosis, angiogenesis	Immune regulation, T helper 2 activation	Matrix remodeling, fibrolysis, angiogenesis, phagocytosis
References	[49]	[50,51]	[52]	[53,54]

### 2.2. Role of Macrophages in Chronic and Inflammatory Diseases

Macrophages play an important role in the development of chronic diseases such as atherosclerosis, rheumatoid arthritis, obesity and cancer [55].

#### 2.2.1. Atherosclerosis (AS)

AS is a chronic inflammatory cardiovascular disease caused by lipid metabolism disorder. Macrophages can absorb low-density lipoproteins via scavenger receptors and hydrolyze them into free cholesterol, part of which is finally converted into cholesteryl ester. The disturbance of the lipid metabolism facilitates the accumulation of macrophages into subendothelium and neointima and their development into foam cells, which secrete proinflammatory factors, such as IL-1β, IL-6 and IL-12 [56,57].

#### 2.2.2. Rheumatoid Arthritis (RA)

RA is a chronic inflammatory autoimmune disease, which occurs with synovial hyperplasia and pannus formation. The pathogenesis of RA is closely related to macrophages, which in the early stage of RA are infiltrated into synovial fluid and polarized into the M1 phenotype. Eventually, inflammation develops due to angiogenesis, the recruitment of T and B cells and the production of cytokines by macrophages, such as TNF-α, IL-6 and IL-1β [58].

#### 2.2.3. Obesity

Obesity is a chronic metabolic disorder which occurs with the excessive accumulation of white adipose tissue (AT) and is related to other diseases [59]. Normally, macrophages located in AT participate in lipolysis, tissue remodeling, immune surveillance and clearing of the cellular debris. The metabolic disorder expands the AT, leading to hypoxia and cell death, which eventually result in excessive leptin and cytokine secretion. Under these conditions, macrophages are recruited into AT and activated via MCP-1, LBT-4 and DAMPs. The production of proinflammatory cytokines by the activated M1-derived macrophages leads to angiogenesis, unchecked basal lipolysis, ectopic lipid storage in other metabolic tissues and insulin resistance [60].

#### 2.2.4. Cancer

One of the factors determining the growth and development of a tumor is the tumor microenvironment (TME), which includes tumor-associated macrophages (TAMs). TAMs are M2-derived tumorigenic macrophages, which promote tumor progression due to their functions. Namely, TAMs secrete growth factors (ornithine, polyamines, EGF, TGF-β, PDGF, and FGF) [61] which stimulate oncogenesis, produce proteolytic enzymes (serine proteases, metalloproteinases, cathepsins, etc.), which contribute to metastasis [62], and encode factors (IL-8, CCL8, bFGF, VEGF, etc.) that promote angiogenesis [63]. Moreover, TAMs impede tumor treatment by producing numerous immunosuppressive molecules, such as HGF, cathepsin and MITF [64]. Therefore, therapeutic strategies targeting TAMs are of great interest, especially as part of synergetic anti-tumor therapies.

### 2.3. Role of Macrophages in Infectious Diseases

#### 2.3.1. Human Immunodeficiency Virus (HIV)

HIV infection occurs with high immune activation and inflammation, caused by high levels of HIV replication, bacterial translocation, coinfection with other viruses and, immune disorders. Macrophages play a crucial role in the innate immune response to pathogens and are recruited to sites of infection and inflammation. Due to their long life time and ability to penetrate different tissues, macrophages have been proposed to play a critical role in the establishment and persistence of the HIV reservoir [65]: the major cellular HIV reservoirs are macrophages and CD4+ T cells, with macrophages being responsible for carrying and spreading the virus [65].

#### 2.3.2. Tuberculosis

The causative bacterium of tuberculosis, mycobacterium tuberculosis (Mtb), is an exquisitely adapted human pathogen which infects via the respiratory route. After being inhaled, Mtb is phagocytized primarily by resident alveolar macrophages, which then can disseminate to other organs. Subsequently, the bacilli either grows unimpeded within host macrophages, resulting in primary progressive disease or reactivation disease after a short period of latency, is killed or adapts to survival within cellular granulomas in a non-replicating state, establishing a latent infection [66]. The ability of Mtb to survive in macrophages without being digested is the main cause of its danger [67]. 

#### 2.3.3. Leishmaniasis 

Leishmania, the causative agent of leishmaniases, is an intracellular parasite of macrophages, transmitted to humans via the bite of its sand fly vector. Different species of Leishmania rely on a range of macrophage receptors in order to detect macrophages and be engulfed [68]. Following phagocytosis, the *Leishmania* promazigotes transform into amazigotes, which then multiply and affect different tissues. The ability to thrive inside macrophages enables *Leishmania* to avoid destruction by the immune system and, eventually, may cause death if the disease is not treated.

## 3. Application of Macrophage-Derived Particles in Therapy

Macrophages are differentiated cells of the immune system that are able to engulf microorganisms, particles and macromolecules. This property of macrophages has attracted attention regarding them as potential carriers of various therapeutic agents, which could allow such advantages as sustained drug release, targeting ability, a prolonged half-life and circulation in the blood, high biocompatibility and low immunoreactivity [69] (Figure 2). 

Interest in this area has increased with an improved understanding of the mechanisms of pathogen recognition by macrophages and their involvement in inflammatory processes due to the presence of PRR receptors and cytokine receptors. In this regard, a large number of studies have been conducted, aimed at developing biomimetic macrophage-mediated systems for selective drug delivery by means of macrophage-derived nanoparticles obtained ex vivo.

### 3.1. Ex Vivo Preparation of Macrophage-Derived Carriers of Therapeutic Agents

In recent years, this concept has been developed in combination with advances in nanotechnology (Figure 3). Since macrophages are able to phagocytize nanoparticles (NPs), therapeutic NPs can be loaded into them ex vivo via simple incubation and then injected into an organism [70,71]. Moreover, in addition to incubation, special methods of introducing therapeutic agents into macrophages, such as hypotonic dialysis [72] and electroporation [73], have been presented. In order to preserve the biological functions of carrier macrophages, which are key to the benefits of a macrophage-mediated drug delivery system, methods of attaching therapeutic agents to the surface of the cell membrane are also used.

In addition to methods of direct use of living cells, methods have been developed for encapsulating therapeutic agents in macrophage membranes or in vesicles derived from macrophages, since they can preserve the biological functions of macrophages necessary for effective drug delivery [74,75,76].

#### 3.1.1. Sources of Macrophages

As part of the use of macrophages in the design of drug delivery systems, researchers proceed either from primary macrophages directly isolated from an animal’s body, or from already-cultured macrophages stored in appropriate banks. Macrophages isolated from the body are usually bone marrow macrophages [77], alveolar macrophages [78] or peritoneal macrophages [79]. Well-known cultured macrophage cell lines are murine cell lines RAW264.7 [80,81,82,83] and J774A.1 [84,85], and human monocytic cell line THP-1 [86,87].

In studies for the direct isolation of BMMs, BALB/c or C57BL/6 house mice are often used, from which bone marrow is isolated and dispersed in an environment containing factors that stimulate the proliferation of monocytes into macrophages (usually cytokines M-CSF, GM-CSF, CSF-1 or IL-3) [88]; this is followed by the isolation of macrophages and their incubation [71,89,90]. In the case of the isolation of peritoneal macrophages, the serous contents of the peritoneum are collected [91,92].

#### 3.1.2. Obtaining of Macrophage-Derived Carriers

In order to avoid the disadvantages of pure drugs, such as immunogenicity, non-selectivity, instability in biological media, a low permeability in tissue, etc., an approach using carrier nanoparticles is often preferred. Thus, many studies in this area are devoted to drug delivery systems involving micelles [93], dendrimers [94] or microgels [95]. These particles are often modified by biomolecules, special ligands or synthetic polymers, for example, PEG [96,97], to avoid the deactivation and degradation of drugs by the mononuclear phagocyte system [3,98]. A different way to increase the therapeutic effectiveness of drugs is to use a cell-mediated delivery system, in particular with the involvement of macrophages. Specific macrophage-mediated systems exhibit several advantages over universal drug delivery systems such as liposomes or albumin NPs due to the inherited macrophage properties. For instance, using macrophages or macrophage-derived vesicles as shells for the particles enables us to improve their tumor capacity in vivo [81] and increase the degree of internalization into tumor cells compared to liposomes [99]. 

Drugs can be loaded directly into living cells (encapsulation), or they can be bound to the outer surface of the cell membrane (adhesion). It is worth noting that the phenotype and “protein profile” of macrophages are crucial factors determining the range of clinical applications of the fabricated macrophage-derived carriers. For instance, M1-like particles are preferable in the development of anti-tumor drug delivery systems, since the abundance of TNF-α [100] and other pro-inflammatory cytokines results in a high tumor accumulation capacity. Moreover, the reproducibility and stability of biomimetic systems are crucial for clinical implementation. Therefore, numerous techniques and approaches have been developed to obtain macrophage-derived particles as drug carriers (Table 2). 

##### Using of Living Cells (Figure 3)

Encapsulation of drugs in macrophages via incubation

The convenience of using macrophages in cell-mediated drug delivery systems is partly due to the fact that drugs, and, in particular, therapeutic nanoparticles, can penetrate macrophages through endocytosis mechanisms (phagocytosis, micropinocytosis, clathrin-mediated endocytosis, caveolar endocytosis) [111]. In this regard, the method of obtaining biomimetic therapeutic agents by incubating macrophages in the presence of drug particles became the first and widespread [71,112,113]. Incubation has good reproducibility due to its simplicity but there are some limitations, namely, the relatively long time of the process and the dependence of the loading efficacy on the physicochemical properties of therapeutic particles. The best results are attained when hydrophobic particles of a medium size are used.

Due to the fact that the drug may have cytotoxicity, therapeutic agents are usually encapsulated in macrophages in the form of drug-loaded particles, which often enables preservation of the biological functions of macrophages. For instance, Choi et al. [102] showed that doxorubicin-loaded liposomes enclosed into mouse peritoneal macrophages via simple incubation exhibited much less toxicity to macrophages than doxorubicin.

In addition to reducing cytotoxicity, loading the drug into special carriers enables one to increase its stability and maintain therapeutic activity in the biological environment. In the assay [113], Batrakova et al. demonstrated that the macrophage-taken nanozime in the form of catalase enclosed in a block ionomer complex showed enzymatic activity after release, while pure catalase lost its activity after incubation with macrophages.

It is known that hydrophobic particles are better engulfed by macrophages [114]. However, the modification of nanocarriers, which increases their hydrophilicity, makes it possible to improve the biocompatibility and stability of the therapeutic agent [115,116]. For example, Madsen et al. [117] utilized gold–silica nanoshells coated with PEG, since PEGylation prevented the aggregation of particles and allowed their efficient encapsulation into macrophages.

b.Encapsulation of drugs in macrophages using hypotonic/resealing method

The osmotic shock of cells through hypotonic dialysis induces cell swelling and the formation of pores. The placing of cells in contact with an appropriate concentration of the substance allows one to load the drug into the macrophage cells using a passive mechanism via pores [118,119]. This method can be applied to create drug delivery systems with macrophages as drug carriers. The application of this method can be limited by the low stability of macrophages under hypotonic conditions, but careful conduction of the procedure specifically in each case can result in a good loading efficacy. For instance, the membrane-impermeable enzyme catalase was packaged into THP-1 cells using this method [72]. It is worth noting that to protect degradation by protease enzymes, encapsulation was carried out in the presence of protease inhibitors; thus, enzymatic activity was preserved.

c.Encapsulation of drugs in macrophage cell membranes using the electroporation/resealing method

Electroporation is a technique enabling to increase the permeability of the cell membrane by applying an electrical field to cells; thus, using this method, drugs, chemicals and biomolecules can be loaded into cells [120], in particular, into macrophages. For example, in the assay used in [73], macrophages were electroporated, and doxorubicin diffused into the cells through the small pores; compared to passive loading, electroporation increased the loading yield of doxorubicin.

d.Adhesion of therapeutic particles to the macrophage membrane (cellular backpacks)

Cellular backpacks are micron-scale patches of a few hundred nanometers in thickness, which can be attached to a cell surface [121,122]. Due to their shape, size and composition, cellular backpacks have the ability to evade engulfment by macrophages, so that both the cell and the drug formulation can be protected from degradation [105]. 

Such microscale structures are usually composed of 4–5 multilayer films, including a payload region and cell attachment region [104,105]. For instance, in the research [104], catalase, a proposed therapeutic agent, was loaded into the payload region of the backpack, which conjugated with macrophages via polyclonal antibodies inserted into the cell attachment region. Most importantly, the attachment of cell backpacks to macrophages did not alter their major functions, including adherence capability and cell activation.

##### Encapsulation of Drugs in Macrophage-Derived Membrane Structures

Encapsulation inside macrophage cellular membranes

The key properties of macrophages, due to which they can be effectively used in therapy as drug carriers, are mainly associated with proteins integrated into their cell membranes. For example, the membrane proteins CD14 and TLR4 bind lipopolysaccharides [123], and CD44 and Mac-1 bind P-selectin and ICAM-1 [76]. Some membrane receptors bind proinflammatory cytokines: CD120 binds tumor necrosis factor (TNF); CD126 and CD130 bind interleukin 6 (IL-6); and CD119 binds interferon-γ (IFN-γ) [74]. In this regard, the isolated cell membrane of macrophages, as well as macrophage-derived vesicles, can serve as the basis of a macrophage-mediated drug delivery system, which enables a high targeting ability and anti-inflammatory properties. 

In order to remove cell contents, the extraction of macrophage membranes is usually carried out using hypo-osmotic swelling, mechanical destruction and several gradient centrifugation steps. In addition, to preserve the structure and activity of transmembrane proteins, protease inhibitors are often added to the medium with macrophages before extraction, and the process itself is carried out at a low temperature [74,106].

In the assay in [74], it was shown that the membrane derivation process not only preserved the transmembrane PRRs (CD14 and TLR4) and receptors of cytokines (CD126, CD130, CD120a, CD120b and CD119), but also resulted in the significant enrichment of these proteins, which enabled high therapeutic efficacy in vivo.

b.Encapsulation inside macrophage-derived vesicles

In order to avoid the difficulties of isolating pure cell membranes of macrophages while preserving the structure and activity of transmembrane proteins, key to the biomimetic parameters of therapeutic particles, macrophage-derived vesicles can be used.

Batrakova et al. [75] attained efficient enzyme incorporation into macrophage-derived vesicles, and the properties of the obtained nanocarriers, including their targeting ability, indicated that key proteins of the macrophage cell membrane were preserved in the vesicles.

Pang et al. [76] developed this approach by using cytochalasin B to stimulate macrophages to produce many microvesicles for nanoparticle cloaking. Analysis proved that the key membrane proteins, such as those involved in self-tolerance (CD45 and CD14) and in adhesion to the inflamed endothelium (CD44, CD18, and Mac-1), were maintained in the obtained microvesicles.

### 3.2. Macrophage-Derived Membranes (or Particles) as Anti-Inflammatory Agents

Inflammation is a complex, local and general protective and adaptive process that occurs in response to pathology or the presence of a pathogen in the body [124]. Monocytes and tissue macrophages circulating in the blood play a major role in the occurrence of inflammation and its course [125]. Important inflammatory mediators are pro-inflammatory cytokines [126], most of which are secreted by M1 macrophages at the site of inflammation and lead to an amplification of the immune response [127]. 

Transmembrane proteins, in particular endotoxin and cytokine receptors, can be preserved in macrophage-derived particles, which means they can be used as therapeutic agents that reduce inflammatory processes by binding inflammatory mediators and endotoxins [74].

Thus, a group of researchers [74] developed a therapeutic detoxification strategy to treat sepsis via a biomimetic macrophage-mediated system; for this purpose, macrophage-derived membranes were used as a coating for polymeric nanoparticles. It was reported that the designed macrophage-derived particles showed the perfect absorption of LPS and cytokines both in vitro and in vivo due to the preservation of key macrophage transmembrane proteins (PRRs and cytokine-receptors) after membrane isolation. The assay indicated that macrophage-derived nanoparticles represent a promising biomimetic detoxification strategy aimed at relieving inflammation by neutralizing endotoxins and LPS.

Using macrophage-derived membranes, Tan et al. [128] demonstrated that they can be effectively used to design therapeutic agents, reducing the level of pro-inflammatory cytokines at the site of inflammation. The obtained nanoparticles inherited the membrane antigenic profile from macrophages and were disguised as mini macrophages to absorb multiple pro-inflammatory substances competitively. It was shown that these macrophage-derived particles can effectively suppress the cytokine-induced activation of macrophages and neutrophils, acting as decoy for cytokines and other inflammatory mediators, thus offering a promising strategy to alleviate inflammatory processes and prevent cytokine storm.

In addition to the anti-inflammatory properties of macrophage-derived particles caused by transmembrane proteins inherited from macrophages, these nanoparticles can be employed as the carriers of drugs. For instance, in the assay [129], a macrophage-mediated system was successfully used to deliver a model drug (atorvastatin) to an atherosclerotic lesion area in mice; this biomimetic approach allowed drug-loaded nanoparticles coated with macrophage membranes to evade MPS and endowed them with targeting ability. Due to the fact that the cooperative binding of cytokines and release of atorvastatin by macrophage-derived nanoparticles decreased the atherosclerotic lesion area, this technique was proposed to be a promising way to treat various inflammatory diseases.

### 3.3. Macrophage-Derived Membranes (or Particles) as Anti-Tumor Agents

At present, anti-tumor drugs often show low efficiency, mainly because of their lack of tumor targeting and their high toxicity to healthy tissues. Since macrophages, as immune antigen-presenting cells, have a long blood half-life and can specifically bind to tumor tissue (Figure 4), applying macrophages in drug delivery can lead to a substantial drug accumulation in tumors (Table 3).

Therapeutic effect obtained from macrophages

In accordance with their functions, the cell membrane of M1 macrophages contains pro-inflammatory cytokines, which lead to the suppression of tumor growth. Moreover, recent studies have indicated that membrane structures derived from M1 macrophages exhibit a better tumor-targeting capacity. Thus, after coating polymeric nanoparticles with macrophage-derived membranes enriched with TNFα, Bhattacharyya and Ghosh [100] showed that the fabricated nanoassemblies triggered apoptosis in cancer cells after treatment in vitro. Therefore, a method of cancer treatment using macrophage-derived nanoparticles of the core–shell type, with a membrane shell isolated from macrophages and containing inflammatory mediators with antiproliferative activity, was proposed.

b.Therapeutic effect due to drug-loaded nanoparticles inside macrophages

In addition to the inflammatory properties of macrophage-derived particles and their protein profile, which determines their anti-cancer activity, they can also serve as carriers of therapeutic agents such as drugs, genes, nanoparticles, etc. Many studies are devoted to the anti-tumor properties of ex vivo obtained macrophages or their membrane structures loaded with a therapeutic agent in the form of drug molecules or nanoparticles. Fu et al. [83] reported that a therapeutically significant amount of doxorubicin (DOX) could be loaded into macrophages without evident cytotoxicity. The DOX-loaded macrophages exhibited tumor-tropic capacity towards 4T1 cancer cells and showed anti-cancer efficacy via tumor suppression, life-span prolongation and metastasis inhibition. 

The loading of drug molecules into macrophages, however, may have a limitation in the form of cytotoxicity. In this regard, the use of nanoparticles is more effective, due to which it is possible to achieve low cytotoxicity for healthy cells, high stability of the therapeutic agent in biological media and the controlled release of the drug. Tao et al. [130] encapsulated polymer nanoparticles loaded with PTX in macrophages and used them in the treatment of glioma. It was shown that nano-PTX-loaded macrophages had a stronger anti-cancer effect on U87-tumor-cells than naked nano-PTX. It is also worth noting that the use of nanoformulations made it possible to reduce the cytotoxicity of PTX for healthy cells and preserve the biological functions of macrophages, thanks to which it is possible to achieve the penetrating ability of macrophage-derived carriers of antitumor drugs. Thus, the ability of DOX-PLGA nanoparticles encapsulated in macrophages to cross the blood–brain barrier and accumulate in glioma region was demonstrated in vivo [137].

It is known that M1 macrophages can selectively accumulate in hypoxic areas of tumors, which are known for their key role in tumor development and resistance to chemotherapy. In their assay [131], Mitragotri et al. demonstrated the tropism of PLGA nanoparticles containing tirapazamine (TPZ) and internalized in macrophages (MAC-TPZ) towards hypoxic regions of 4T1 tumors; eventually it resulted in a 3.7-fold greater reduction in tumor weight compared to TPZ alone.

c.Therapeutic effect due to surface engineering of macrophages

Therapeutic agents encapsulated in macrophages can be exposed to phagosomes, which causes restrictions on the utilization of macrophages as drug carriers; a drug can affect the biological functions of macrophages, and its isolation in phagosomes can lead to its degradation and a reduced drug release rate. For this reason, phagocytosis-resistant drug-loaded backpacks that are able to attach to the outer membrane of macrophages have been developed [105,138,139,140]. There are known methods of modifying the cell membranes of macrophages with other particles, for example, quantum dots and dendrimers [141]. Sugimoto et al. [133] showed that the surface modification of macrophages with nucleic acid aptamers improved the capture of T-cell acute lymphoblastic leukemia cells and enhanced their anticancer immune response.

d.Therapeutic effect due to bioengineered species

In addition to the use of nanoparticles, bioengineered organisms capable of having an anti-cancer effect can also be loaded into macrophages. For instance, in their assay [101], researchers showed that the macrophage-mediated tumor-targeted delivery of modified bacteria VNP20009 substantially suppressed melanoma in mice. Muthana et al. [134] utilized macrophages as carriers of oncolytic adenovirus, which effectively accumulated in hypoxic tumor areas, inhibited tumor growth and reduced pulmonary metastases of prostate cancer in mice.

e.Photothermal therapy

Macrophage-derived particles can also be used in photothermal cancer therapy, in which a nanomaterial with a high photothermal conversion efficiency is injected into the body. Such nanomaterials, when used by themselves, often accumulate in healthy tissues and organs, which is why they can cause long-term harmful effects [142]. Encapsulating such materials into macrophage membranes can solve this problem and lead to drug delivery systems with a good photothermal conversion ability, biocompatibility, ability to escape immune responses, and ability to target tumors. In many studies, this has been proven by the example of Au nanorods (AuNRs) encapsulated in macrophages [143]. Moreover, photothermal therapy can be carried out in conjunction with other types of anti-cancer therapy; for instance, it is reported that the joint encapsulation of DOX loaded into temperature-sensitive liposomes and AuNRs into macrophages results in synergetic chemo–photothermal therapy, enabling the researchers to target and kill tumor cells in vivo [81].

### 3.4. Macrophage-Derived Membranes (or Particles) for the Treatment of Infectious Diseases

Macrophages mediate a wide range of infectious diseases. They play a key role in protecting the body against many pathogens, including viruses, bacteria and parasites. Due to the functional activity of the transmembrane proteins of macrophages, macrophage-derived particles can serve as targeting carriers of antimicrobial agents.

Treatment of viral infections

In the already-mentioned study [128], in addition to alleviating inflammation by absorbing inflammatory mediators, polymer nanoparticles wrapped in a macrophage membrane reduced virus replication, thereby increasing the survival rate among mice infected with the SARS-CoV-2 model. It was shown that the membrane shell of the particles contained the receptor ACE II, which is essential for SARS-CoV-2 targeting.

Due to the ability of macrophages to pass through the blood–brain barrier, macrophage-derived particles loaded with antiviral drugs can be used for the antiviral therapy of neurological complications of AIDS. For instance, macrophages loaded with indinavir were used to treat mice with HIV-1 encephalitis (HIV), and the study revealed a steady accumulation of the drug and a decrease in HIV-1 replication in HIVE brain regions [144].

b.Wound healing and treatment of bacterial infections

Macrophage membranes, which contain TLRs, can be used for developing targeted drug delivery systems to bacteria. In this regard, before wrapping therapeutic agents in the membrane, macrophages are often pre-exposed to a pathogen in order to enrich the membrane with the necessary proteins.

Using hydrogels or particles surrounded by a macrophage membrane, it is possible to increase the effectiveness of the healing of bacteria-infected wounds with photothermal therapy. Liu et al. [145] developed AuNR-containing hydrogel coated with a bacteria-pretreated macrophage membrane. The resultant hydrogels specifically detected the bacteria source, and destroyed 98% bacteria in vitro under NIR irradiation. Moreover, the hydrogels implanted on the dorsal area of rats could facilitate healing of the infected wound and avoid secondary damage during peeling. Similarly, Zhang et al. [146] showed that applying a pretreated macrophage membrane to the surface of gold–silver nanoparticles increased their targeting ability and prolonged their blood circulation time. The developed membrane-coated nanoparticles also were proposed as potential drug delivery vehicles, so that photothermal therapy could be applied synergistically with drug therapy.

Due to their affinity to inflammatory mediators and ability to bear drugs, macrophage-derived particles have high therapeutic potential in the treatment of inflammatory diseases, such as atherosclerosis and sepsis. For instance, to simultaneously realize the pathogen elimination and inflammation resolution in a region infected with a pathogen in periodontitis, Xu et al. [147] utilized simvastatin-loaded nanoparticles coated with pathogen-pretreated macrophage membranes, which simultaneously diminished the atheromatous plaque formation in atherosclerosis and rejuvenated the alveolar bone loss in periodontitis. In a study on the treatment of sepsis [148], macrophage membrane-coated antimicrobial peptide nanoparticles effectively delivered a drug and were retained at the site of infection, eventually decreasing the level of inflammatory factors and increasing the survival rate.

## 4. Macrophage-Mediated Therapy via Macrophage Targeting

### 4.1. Design of Therapeutic Agents Targeting Macrophages 

Macrophages mediate the pathological processes of inflammatory diseases, including oncology and infectious diseases. Since macrophages are closely related to tumor development, as well as due to the existence of pathogens acting through macrophages, studies aimed at the design of drug-loaded micro- and nanoparticles targeting macrophages are of great interest. 

Therapeutic agents in the form of drug-loaded particles can be delivered to particular organs or cells based on their physicochemical properties, such as their size, shape, charge and solubility (passive targeting), or can be delivered to macrophages via specific targeting ligands (active targeting).

#### 4.1.1. Passive Macrophage-Targeting Therapeutic Agents

Passive delivery is based on the pharmacokinetics of NPs, the enhanced permeability and retention (EPR) effect, and immune responses of the targeted tissue, leading to the accumulation of NPs. The efficiency of capture of particles by macrophages as part of a passive delivery strategy is affected by their following parameters: size, shape, surface charge and hydrophilicity.

Size

The size of the particles can affect the efficiency of their capture by macrophages, although the cellular uptake depends on environmental conditions. For instance, the uptake of non-modified liposomes by rat alveolar macrophages in vitro increased with an increase in particle size and became constant at 1000 nm, while the uptake of non-modified liposomes by alveolar macrophages in vivo increased with an increase in particle size in the range 100–2000 nm, which was due to the opsonization by lung surfactant proteins [149]. In addition, it is reported that the size of the particles affects the phagocytic capacity, endocytosis speed and endocytosis mechanism of the cell [150,151]. 

It is also known that bio-distribution in RES organs is affected by particle size. For example, NPs up to 500 nm in size accumulate in the liver and lungs; NPs of 10–300 nm in size accumulate predominantly in the liver and spleen; and NPs of 1–20 nm in size are usually degraded by macrophages in the kidneys [152,153]. In addition, epithelial destruction and vascular leakage occur in areas of inflammation and in solid tumors [124,154]. Therefore, NPs with a proper size can preferentially extravasate from the blood into the interstitial spaces and accumulate in inflammation sites or tumor tissues via the EPR effect [155,156].

b.Shape

Particle shape has a significant impact on macrophage cellular uptake and can be exploited for controlling the efficacy of drug delivery to macrophages. Smith et al. [157] proved that particle shape independently influences binding and internalization by macrophages. Interestingly, they found that the attachment of particles to macrophages could be ranked in the following order: prolate ellipsoids > oblate ellipsoids > spheres. However, the internalization of particles followed a different rank: oblate ellipsoids >> spheres > prolate ellipsoids. The effect of the particle shape can be explained by the fact that endocytosis is an actin-dependent process, and, therefore, the internalization of particles with a larger aspect ratio requires more energy to perform the cytoskeleton remodeling [153,157,158].

c.Surface charge and hydrophilicity

Surface charge is another factor that influences macrophage uptake, and many assays both in vitro and in vivo have indicated that charged particles are more likely to be taken up by macrophages than neutral particles. Despite the fact that the absorption of positively charged particles by cells is usually easier as a result of electrostatic interactions [159], it has been shown that the same increase in cellular uptake by macrophages can be achieved with an increase in both the negative and positive charge of particles [159,160,161]. However, the role of charge on macrophage uptake is still controversial, with contradictory observations in the literature [162]. 

Hydrophilicity is another parameter that strongly affects the capture of particles by macrophages in vivo. Hydrophilicity, as well as surface charge, can impact the adsorption of opsonin, thus influencing the uptake of NPs by macrophages. Increased hydrophilicity results in a lower degree of protein adsorption and reduced uptake by macrophages [163,164,165]. It is often used to hide NPs from MPS by covering them with PEG [166].

#### 4.1.2. Active Macrophage-Targeting Therapeutic Agents

Active targeting can significantly enhance the selectivity of macrophage-mediated therapy due to specific interactions between the therapeutic agent and the cell. This approach involves the direct use of the agonists or antagonists of macrophage receptors, or modification of the surface of NPs with ligands or an antigen in order to establish selective interaction with macrophage receptors. Many studies demonstrate the advantages and therapeutic potential of the active targeting of macrophages in the treatment of oncological and infectious diseases. In this regard, promising results can be attained by using therapeutic agents specifically delivering drugs to macrophages (Table 4).

Below are the main approaches to active macrophage targeting based on the interaction of therapeutic agents with different macrophage receptors (Figure 5).

Toll-like receptor targeting

Toll-like receptors (TLRs) are well-defined pattern recognition receptors responsible for pathogen recognition and the induction of innate immune responses via signaling pathways. TLRs can detect various endogenous damage-associated molecular patterns (DAMPs) and pathogen-associated molecular patterns (PAMPs) [185]. The activation of TLRs initiate a variety of downstream signaling cascades and signaling pathways, leading to the production of inflammatory cytokines or type I IFNs [186]. The activation of TLR signaling is also crucial to the induction of antigen-specific adaptive immune responses by activating the adaptive immune cells for the clearance of invading pathogens [185].

Among the functional TLRs identified in humans, some are localized on the cell surface (TLR1, TLR2, TLR4, and TLR5) and others in intracellular compartments (TLR3, TLR7, TLR8, and TLR9) [185]. Cell-surface TLRs mainly detect membrane components of the pathogens such as proteins, lipoproteins, lipids and lipopolysaccharides (LPS), while intracellular TLRs mainly recognize nucleic acids derived from pathogens or self-nucleic acids in a pathological condition [185].

Since TLRs are involved in the production of pro-inflammatory mediators and the activation of immune responses, TLRs present an attractive target for the more precise manipulation of the function of macrophages [187]. Recent studies have demonstrated that TLR pathways play a significant role in polarizing macrophages. Therefore, TLRs can serve as a target for modeling a macrophage’s phenotype, for example, as part of tumor treatment via TAM reprogramming [188]. In addition, TLR ligands have found application in the context of infectious, inflammatory and autoimmune diseases [187].

b.Scavenger receptor targeting

Scavenger receptors (SRs) are a diverse superfamily of cell surface receptors. They are expressed by myeloid cells (macrophages and dendritic cells) and certain endothelial cells. Being a subclass of the membrane-bound pattern recognition receptors (PRRs), they play an important role in the cellular uptake and clearance of endogenous host molecules and apoptotic cells, and exogenous components marked with pathogen-associated molecular patterns (PAMPs) [189]. Removal is often carried out by simple endocytosis but might entail more complex processes, such as micropinocytosis or phagocytosis, which both require elaborate signal transduction [189,190]. 

Due to the fact that SRs are involved in phagocytosis and in endocytosis, these receptors are potential intermediaries in macrophage-targeting therapy, which can facilitate the selective delivery of therapeutic agents into macrophages. C-type lectin receptors (CLRs), which recognize conserved carbohydrate structures, attract a lot of attention [191]. In this regard, many assays are devoted to developing drug nanocarriers targeting mannose receptor (also known as CD206) [192,193]. Thus, mannosylated therapeutic agents are of great interest and many researchers have shown that a high targeting ability can be achieved via the modification of drug nanocarriers with mannose [194,195,196]. Other pathogen-associated components that are used for SR-mediated targeted delivery are galactose [197,198,199], dextran and its derivatives [200,201,202]. Selective delivery can also be achieved by encapsulating therapeutic agents in glucan particles derived from the yeast cell wall, which can be recognized by CLR Dectin-1 [174,175]. In addition, bio-nanocapsules (BNCs) derived from pathogens, such as virus envelope particles [203] or bacteria-like particles [204,205], can be directly used as macrophage-targeted drug carriers.

c.Fc-receptor targeting

So-called Fc-receptors (FcRs) for different immunoglobulin isotypes (IgA, IgE, IgM, and IgG) are involved in regulating and executing antibody-mediated responses [206]. FcRs are widely expressed on cells of the immune system, including macrophages [206]. These receptors recognize antibodies that are attached to infected cells or invading pathogens and stimulate phagocytosis and endocytosis [207]. 

Since FcR activation stimulates phagocytosis and endocytosis, FcR-mediated drug delivery strategies targeting macrophages have been developed. In this regard, tuftsin—a tetrapeptide formed by the enzymatic cleavage of the Fc portion of the immunoglobulin (IgG) molecule—has gained a lot of attention due to its ability to activate FcR [176,208,209,210]. For instance, Jain et al. [176] developed tuftsin-modified NPs and noted a much higher cellular uptake by macrophages in vitro than non-modified or scrambled peptide-modified NPs. In addition, a tuftsin derivative, tuftsin tetramer, can dramatically enhance uptake into macrophages [211].

d.Targeting of other receptors

In addition to the above-mentioned important receptors expressed by macrophages, the folate receptors [212] and CD44 [213] are considered to be potential mediators in macrophage-targeting therapy strategies. 

Folate receptors are expressed on the surface of activated macrophages, known to be upregulated in the macrophages in rheumatoid arthritis and pulmonary fibrosis [214]. A high macrophage-targeting ability can be achieved by the modification of therapeutic agents with folic acid. Thus, folate-conjugated particles, such as dendrimers [178], chitosan NPs [215], liposomes [216] and human serum albumin NPs [179] exhibited enhanced macrophage uptake when compared with non-folated particles.

CD44 is a receptor for hyaluronic acid-mediated motility (RHAMM). Nanoparticles modified with hyaluronic acid (HA) can be recognized by CD44 and be taken by macrophages [213]. In recent assays, such modification of drug-loaded nanoparticles, such as micelles [180], liposomes [181] and polymeric NPs [182], enabled efficient cellular uptake by macrophages.

### 4.2. Macrophage Targeting in Anti-Inflammation Therapy

Since macrophages play an indispensable role in initiating and developing inflammatory processes, they are a potential target in anti-inflammatory therapy. Important factors which promote macrophage recruitment into the site of inflammation are cell adhesion molecules, such as ICAM-1 and VCAM-1, the inhibition of which can suppress inflammation. For instance, Sager et al. [217] showed that silencing endothelial cell adhesion molecules using siRNA reduced monocyte recruitment into atherosclerotic lesions.

Anti-inflammatory cytokines, such as IL-10, and anti-inflammatory drugs can also be applied to suppress inflammation. Thus, IL-10 delivered by polymeric nanocarriers was bioactive and reduced the production of pro-inflammatory cytokine IL-1β in the atherosclerotic lesion and led to significant regression in the plaque size [218]. Local inflammation treatment based on mannose-modified nanoparticles loaded with anti-inflammatory diclofenac was successfully applied to wound healing [219]; drug-loaded macrophage-targeted nanoparticles showed an enhanced anti-inflammatory effect in wound healing compared to the drug-coating-free suture.

The overactivation of TLRs leads to the production of high levels of IFN and other cytokines, which can cause chronic inflammation [220]. Moreover, the chronic activation of TLRs via interaction with DAMPs may stimulate T- and B-cells responses and contribute to the development of autoimmunity. Therefore, TLR antagonists, such as TLR2 antagonists AT1-AT8 [221], have been proposed as agents to attenuate inflammation. 

### 4.3. Macrophage Targeting in Anti-Tumor Therapy

In the case of oncological diseases, due to the influence of macrophages on tumor development, macrophages are the preferred target for various therapeutic agents. On the one hand, M1 macrophages inhibit tumor growth and metastasis; on the other hand, TAMs (M2 macrophages) provoke tumor growth and angiogenesis. Therefore, a promising strategy is to increase the ratio of M1 macrophages/TAMs at the tumor site, which can be achieved by inhibiting macrophage recruitment, the direct depletion of TAMs, blocking “don’t eat me” signals, and reprogramming TAMs. In this regard, many strategies have been proposed, including blocking the CCL2/CCR2 axis [222] and the CD47-SIRPα pathway [223].

CCR2 is predominantly expressed by monocytes/macrophages with strong proinflammatory functions. CCR2 is a CC chemokine receptor for monocyte chemoattractant protein-1 (CCL2), which is involved in macrophage recruitment. In this regard, in order to inhibit TAM recruitment, CCR2 antagonists, such as RS102895 [224], BMS CCR2 22 (Tocris) [225] and CCX872 [226], can be used. For instance, CCX872 has exhibited good inhibition of macrophage recruitment due to its high affinity to CCR2.

Signal regulatory protein α (SIRPα) is a regulatory membrane glycoprotein from the SIRP family expressed mainly by myeloid cells (macrophages, monocytes, granulocytes, and myeloid dendritic cells) [227]. SIRPα acts as an inhibitory receptor and interacts with a broadly expressed transmembrane protein, CD47, also called the “don’t eat me” signal, which inhibits phagocytosis [228]. Therefore, SIRPα inhibitors, such as CD47 analogues or anti-SIRPα antibodies [229,230], are proposed as therapeutic agents that promote the phagocytosis of tumor cells by macrophages. For example, the monoclonal antibody KWAR23, which binds human SIRPα with high affinity and disrupts its binding to CD47, has been shown to be a promising candidate in therapy, though in combination with tumor-opsonizing monoclonal antibodies [231].

In addition, the TAM-targeted delivery of therapeutic agents is another promising strategy. In this regard, Siglec, cell surface receptors that bind sialic acid (SA), are a potential target. Among the Siglec family receptors, the SA adhesion protein Siglec-1 is one of the most abundant superficial receptors of TAMs and can mediate endocytosis after binding to SA. Recently, the modification of drug-loaded liposomes [183,184] with sialic acid has enabled a high targeting ability of drug nanocarriers. For instance, Deng et al. [232] demonstrated that the cellular uptake of liposomes modified with a sialic acid–cholesterol conjugate was increased compared with other formulations.

Inhibition of macrophage recruitment

Biomolecules recruiting monocytes, such as VEGF, CSF-1, CCL2 and CCL5, are involved in macrophage recruitment to the tumor area and, as a result, in increasing the number of TAMs. Inhibitors of these chemoattractants and their receptors can suppress macrophage recruitment and monocyte proliferation in TAMs, thereby reducing tumor growth and dissemination [233].

In this regard, the CCL2/CCR2 axis attracts a lot of attention; it is known that its blocking is an effective approach to inhibit macrophage recruitment in tumor sites [234]. This can be implemented via anti-CCL2 therapy or anti-CCR2 therapy.

Loberg et al. [235] used monoclonal antibodies C1142, specifically binding to CCL2, and with their help successfully inhibited the growth and metastasis of a tumor by blocking the infiltration of TAMs. Similarly, CANTO888 antibodies were used for anti-CCL2 inhibition [236]; despite the fact that in vivo docetaxel cancer treatment was more effective than CANTO888 antibody treatment alone, the inhibition of CCL2 in combination with docetaxel significantly reduced the tumor load and induced tumor regression.

Inhibiting CCR2 is another potent therapeutic strategy, which can be implemented by anti-CCR2 antibodies, such as CCX872 [219]. An approach to blocking the CCL2/CCR2 axis through the inhibition of mRNA translation is shown in the research [237]. Wang et al. used cationic nanoparticles for targeted delivery of CCR2siRNA. Due to the charge modification, the particles were effectively engulfed by monocytes, due to which an efficient inhibition of macrophage recruitment and TAM infiltration was achieved.

b.Targeting Anti-Phagocytic Checkpoints

Being important immune cells, macrophages are able to engulf tumor cells and present tumor-specific antigens to induce adaptive immunity. However, tumor cells can evade phagocytosis by macrophages due to the high expression of “don’t eat me” signals [238]. 

Among “don’t eat me” signals, CD47 is the most studied antiphagocytic signal, and it is known that it prevents phagocytosis through interaction with the SIRPα integrated into the macrophage membrane [239]. Blocking the CD47-SIRPα pathway via anti-CD47 therapy or anti-SIRPα therapy is a way to restore the antitumor activity of TAMs [240]. 

Chao et al. [241] demonstrated that a blocking monoclonal antibody against CD47 enabled the phagocytosis of acute lymphoblastic leukemia (ALL) cells by macrophages in vitro and inhibited tumor engraftment in vivo. Moreover, anti-CD47 antibody eliminated ALL in the peripheral blood, bone marrow, spleen, and liver of mice engrafted with primary human ALL. It has been shown that in addition to antibodies, SIRPα analogues can be used to neutralize CD47. For instance, Koh et al. [242] utilized exosomes containing SIRPα variants, which induced significantly enhanced tumor phagocytosis and primed cells for an effective anti-tumor T cell response. Encouraging results can be achieved with the CD47 inhibitor magrolimab, the ongoing phase 2 trial of which is evaluating its tolerability, safety and effectiveness in the treatment of myeloma, especially in combination with other anti-cancer therapies [243].

CD47 is expressed in all types of cells, while SIRPα is only expressed on the surface of myeloid cells (macrophages, monocytes, granulocytes, and myeloid dendritic cells). Therefore, in some cases anti-SIRPα therapy is preferable [244]. In this regard, monoclonal antibodies that bind SIRPα with high affinity can be used. For instance, Ring et al. [226] showed that the anti-SIRPα antibody KWAR23 in combination with tumor-opsonizing monoclonal antibodies greatly augmented the myeloid cell-dependent killing of human tumor-derived cell lines in vitro and in vivo.

c.TAM depletion

TAM depletion is another approach to macrophage-targeted therapy that can help reduce angiogenesis, reactivate immune surveillance, and ultimately suppress tumor growth. For this purpose, various anti-cancer drugs or colony stimulating factor inhibitors, such as CSF-1, can be used.

Since TAMs cause immunosuppression via inhibiting the recruitment of T cells through cytokines, superficial immune checkpoint ligands, and exosomes, the application of immunomodulatory drugs, such as anti-programmed cell death 1 (PD-1) or anti-PD-ligand 1 drugs, is limited. Therefore, in order to improve anti-PD-1/PD-L1 therapy, TAM depletion as part of the synergetic therapy is effective in inhibiting tumor growth [245,246].

Diphtheria toxin treatment during tumor initiation or the depletion of TAMs in established tumors prevented pancreatic cancer initiation [247]; in the case of pre-established tumors, TAM depletion inhibited tumor growth and, in some cases, induced tumor regression.

Drug-carrying nanoparticles modified for targeted delivery to macrophages can also be effectively used to deplete TAMs. Zhou et al. [227] utilized sialic acid–cholesterol-conjugate modified liposomes loaded with epirubicin (EPI-SAL) and showed that EPI-SAL achieved enhanced accumulation of the drug into TAMs; the antitumor studies indicated that EPI-SAL provided strong antitumor activity by modulating the tumor microenvironment with the depletion of TAMs. Another anti-cancer drug, doxorubicin, was loaded into liposomes modified with PEG-D-mannose and PEG-L-fucose conjugates as macrophage receptor ligands [248]; the dual-ligand modified PEGylated liposomes achieved an increased distribution of DOX in tumor tissues and the superior tumor inhibitory rate via modulation of the tumor microenvironment with the exhaustion of TAMs was shown.

d.Reprogramming of TAMs

Under the influence of various factors, TAMs can switch their phenotype between tumoricidal M1- and protumorigenic M2 macrophages, which is inspiring the design of therapeutic agents targeting this macrophage plasticity. Thus, one of the promising immunotherapeutic strategies for cancer therapy is the repolarization (reprogramming) of TAMs towards an anti-tumor M1 phenotype [249]. Drugs, cytokines, immunoagonists, CpG oligonucleatids, siRNA and ROS/O2-generating nanoparticles can be used to reprogram TAMs.

In many studies, liposomes are used to encapsulate a drug and its targeted delivery to macrophages. Sousa et al. [250] showed that the effect of liposome-encapsulated zoledronate on macrophages cultured in a conditioned environment of breast cancer cells increased the content of markers of the M1 phenotype of macrophages (iNOS and TNF-α). Later, Tan et al. [184] designed liposomes modified with sialic acid (SL) and loaded zoledronic acid (ZA) into them; thanks to the modification, these drug nanocarriers (ZA-SL) could effectively deliver ZA to TAMs. In vivo experiments showed that ZA-SL cancer treatment increased the M1/M2 ratio, which was partly caused by the phenotypic remodeling of M2-like TAMs. Studies have shown that ZA reverses the polarity of TAMs from M2-like to M1-like by attenuating IL-10, VEGF, and MMP-9 production and recovering iNOS expression [251].

Polymer-based nanoparticles can also be used as nanocarriers of drugs targeted at macrophages for TAM repolarization. Wang et al. [252] developed poly β-amino ester-based NPs that could adapt by the systemic administration and release of IL-12 in the tumor microenvironment, subsequently re-educating TAMs. The nanocarriers loaded with IL-12 exhibited enhanced tumor accumulation, and extended the circulation half-life and therapeutic efficacy of encapsulated IL-12 compared to free IL-12. Cheng et al. [253] proposed a multifunctional macrophage targeting system to deliver CpG oligodeoxynucleotides to macrophages; they used mannosylated carboxymethyl chitosan/protamine sulfate/CaCO3/CpG nanoparticles, which were efficiently taken by macrophages and exerted a polarizing effect on them, increasing the production of proinflammatory cytokines including IL-12, IL-6, and TNFα.

Downregulation of CSF-1R is known to reprogram the immunosuppressive M2 macrophages to the immunostimulatory phenotype, M1 macrophages. Sialic acid-targeted cyclodextrin-based nanoparticles were developed to deliver CSF-1R siRNA to TAMs [254]; in in vitro experiments, the nanoparticles achieved cell-specific delivery to TAMs, eventually polarizing M2-like TAMs to an M1 phenotype, which enhanced the level of apoptosis in the prostate cancer cells.

Since reactive oxygen species (ROS) are important modulators of macrophage activation and polarization towards a tumoricidal M1 phenotype, ROS-generating NPs can be used as therapeutic agents modulating the tumor microenvironment. Nascimento et al. [255], using breast cancer models in vitro, found that polyaniline-coated maghemite (γ-Fe_2_O_3_) nanoparticles could be easily taken by M2-like macrophages and could re-educate them towards a pro-inflammatory profile via ROS generation. Immunotherapy can be enhanced by tumor-derived antigenic microparticles loaded with nano-Fe_3_O_4_- and CpG-containing liposomes, which can repolarize TAMs to M1 macrophages and induce the infiltration of cytotoxic T lymphocytes at the tumor site [256].

Additionally, due to TAM recruitment driven by hypoxia and the accumulation of TAMs in hypoxic regions of solid tumors, oxygen-generating NPs can regulate TAM repolarization by reducing hypoxia. Thus, Youn et al. [257] developed mannose-decorated/macrophage membrane-coated upconverting nanoparticles that contained particles generating ROS and oxygen under light irradiation.

### 4.4. Macrophage-Targeting in the Therapy of Infectious Diseases 

Macrophages, as crucial components of immune system, can engulf and digest microbes. However, some pathogenic microorganisms have the ability to survive the digestion and utilize macrophages as reservoirs for safe haven, avoiding the action of other cells of the host immune system [258,259,260]. These microorganisms can circumvent the effectiveness of antibiotics by surviving inside host macrophages. Since therapeutics in current use have varying abilities to enter macrophages, there is an interest in using special drug delivery systems via drug-loaded nanoparticles to treat intracellular infections. The targeted delivery of drugs to macrophages is considered below on the example of HIV, tuberculosis and leishmaniasis.

Viral infectious diseases

Human immunodeficiency virus (HIV) is a lentivirus that leads to acquired immunodeficiency syndrome (AIDS), an immunocompromised condition that increases susceptibility to macrophage resident diseases. Since the major cellular HIV reservoirs are macrophages and CD4+ T cells, with macrophages being responsible for carrying and spreading the virus, the development of methods for the direct delivery of anti-HIV drugs to macrophages is of great interest [65,261].

Recently, drug-loaded nanoparticles, such as liposomes, polymer NPs and dendrimers, have been considered as potential therapeutic agents for treating HIV. In order to achieve a high targeting ability, Jain et al. [262] conjugated efavirenz-loaded dendrimers with tuftsin; the obtained therapeutic agents not only exhibited excellent cellular uptake but also possessed relatively low cytotoxicity with simultaneous high antiviral activity. Similarly, Jain et al. [263] developed stavudine-loaded mannosylated liposomes, which also exhibited a high targeting ability and increased biocompatibility in comparison with the pure drug. The study of the kinetics of release, the effectiveness of loading antiviral drugs in polymeric nanoparticles and their targeting ability revealed their potential as anti-HIV drug carriers [264,265,266]. Thus, Krishnan et al. [267], using chitosan carriers loaded with saquinavir, demonstrated a drug encapsulation efficiency of 75% and cell targeting efficiency greater than 92%; the saquinavir-loaded chitosan carriers exhibited superior control of the viral proliferation compared to the control drug.

b.Tuberculosis

Tuberculosis (TB) is a lung infection caused by Mycobacterium tuberculosis (Mtb). Mtb primarily infects host macrophages, developing special survival and reproduction strategies in these highly specialized cells [268]. Most of the known anti-tubercular agents are less effective in vivo due to the low macrophage permeability and rapid degradation of these drugs [269]. The use of antibiotic-loaded, macrophage-targeted nanoparticles enables a prolonged and systemic dose of anti-tubercular antibiotics [269].

Many assays are devoted to the development of nanoparticles for active targeted delivery, for example, by modification with mannose [192,193]. For instance, Huang et al. [194] designed mannose-modified solid lipid nanoparticles (SLNs) containing the pH-sensitive prodrug of isoniazid (INH) for the treatment of latent tuberculosis infection. A fourfold increase in intracellular antibiotic efficacy and enhanced macrophage uptake in vitro was observed, while in vivo assays showed that the level of the colony-forming unit was decreased in the SLN group compared to the free INH group. Later, in order to design macrophage-targeted delivery in TB, Ambrus et al. [270] developed nanomediated isoniazid-loaded dry powder for inhalation, based on mannosylated chitosan and hyaluronic acid hybrid nanoparticles, which were found to be a promising vehicle for targeting TB-infected macrophages. Pi et al. [170] first reported the bactericidal effects of selenium nanoparticles and introduced a novel nanomaterial-assisted anti-TB strategy manipulating isoniazid-incorporated mannosylated selenium (Ison@Man-Se) NPs for synergistic drug killing and phagolysosomal destructions of Mtb. They found that Ison@Man-Se NPs selectively entered macrophages and accumulated in lysosomes, releasing isoniazid. Furthermore, Ison@Man-Se/Man-Se NPs could trigger the fusion of Mtb into lysosomes, so that the synergistic lysosomal and isoniazid destruction of Mtb was achieved.

c.Protozoan infectious diseases

Leishmaniasis is a wide array of clinical manifestations caused by leishmania, a parasitic protozoan [271]. The intracellular localization of leishmania inside the phagolysosome of host macrophages limits chemotherapy treatment. In addition, the use of antileishmanial drugs is often compromised because of their toxicity and limited bioavailability [272]. Macrophage-targeted therapeutic agents can solve these problems [272].

Compounds of pentavalent antimony, such as sodium stibogluconate (SSG), used in the treatment of leishmaniasis have high toxicity; encapsulation of the drug in nanocarriers can help to overcome this disadvantage. For instance, Khan et al. [273] developed nano-deformable liposomes (NDLs) for the dermal delivery of SSG against cutaneous leishmaniasis; compared with the pure drug solution, NDLs displayed an increase in the selectivity index, a decrease in the cytotoxicity and a higher anti-leishmanial activity, due to effective healing of the lesion and a successful reduction in the parasitic load in vivo. Recently, targeting nanoparticles loaded with other antileishmanial drugs, such as amphotericin B (AmB) and paromomycin (PM), have also been utilized for the treatment of leishmaniasis. Heli et al. [274] investigated the effect of ligand modification of PM-loaded chitosan NPs on their anti-leishmanial activity and found the mannosylated formulation to be a suitable targeted drug delivery system for uptake into Leishmania-infected macrophages without any cytotoxic activity. Similarly, Das et al. [275] prepared a mannose containing composite hydrogel loaded with AmB and showed it to be a suitable candidate for the treatment of leishmaniasis due to its injectability, biodegradability, non-cytotoxicity and efficient drug delivery properties.

### 4.5. Potency of Macrophage Targeting via CD206 Receptor 

Targeted delivery to macrophages (including targeting to CD206 or Siglec-1 receptors) opens up numerous opportunities to influence a wide range of diseases and pathological conditions, of which they are the driver or direct participant. These are infectious diseases that have the property of forming a reservoir of latent forms inside macrophages (tuberculosis, HIV, Ebola, etc.); a number of oncological diseases where macrophages make the main contribution to the creation of an immunosuppressive microenvironment of tumors, making them “cold”, i.e., practically invisible to the immune system; and autoimmune diseases (rheumatoid arthritis, osteoarthritis, multiple sclerosis), the pathogenesis of which is associated, on the contrary, with the excessive pro-inflammatory activity of macrophages.

The use of mannosylated drug delivery systems using the idea of biomimetics for oligosaccharidic patterns of microorganisms to target macrophage mannose receptors has been recently studied in detail by our scientific group in a series of papers [192,276,277,278,279,280] (Figure 6), in which we aimed to create a targeted drug delivery system for the treatment of a number of infectious, inflammation and oncological diseases. On the basis of ligands specific to the macrophage receptor CD206, a system of targeted delivery of therapeutic “cargo”, enhanced with adjuvants (showing a synergistic effect with the main drug), into macrophages was developed [192,278,280,281,282,283,284,285,286,287] (Figure 6 shows a macrophage with its receptors recognizing mannosylated polymers). The use of a macrophage CD206 receptor as a target provides a high selectivity for drug delivery and does not require the use of immune-active compounds (interleukins, proteins, microRNAs, etc.). Systematic studies of the ligands of the CD206 receptor allowed us to develop a series of specific molecular containers of different molecular architectures, carrying an oligomannoside ligand of a complex structure with optimal affinity to the mannose receptors of macrophages. We modeled the interaction of CD206 with more than a hundred relevant carbohydrate structures [276,288]; about two dozen of them were studied experimentally. As a result, optimized polymeric ligands were developed based on polyethyleneimine, mannan, and chitosan grafted with cyclodextrins (Figure 6—center) and provided the effect of accumulation of a therapeutic “cargo” in macrophages, which significantly increased organ bioavailability (and the accumulation of drugs in the lungs), and the permeability of bacterial cells to drugs was developed.

In addition to the optimized carbohydrate ligand providing binding to the surface receptor, the delivery system should provide a high degree of loading of therapeutic “cargo” and adjuvant, and their stimulus-sensitive release inside cells. Stimulus-sensitive release is achieved through the use of a polymer forming a compact nanoparticle at neutral pH, and unfolding at a low pH inside the bacterial endosome. A high degree of loading is achieved by grafting this polymer with cyclodextrin molecules that are able to bind therapeutic cargo (antibiotic), as well as synergistically acting adjuvants (which are otherwise poorly soluble and do not have the necessary bioavailability parameters). As a result, each polymer molecule is able to carry up to 20 “cargo” molecules (80% loading rate), which are approximately equally divided between the antibiotic and the adjuvant, and deliver them inside macrophages due to the binding of the carbohydrate ligand to CD206.

A significant increase in efficiency can be achieved by using adjuvants, which enhance the effect of an antibiotic by inhibiting efflux and increasing the permeability of the bacterial membrane [280,284,285]. Currently, the direction of biocompatible medicine is actively developing, in other words, the use of safe natural extracts and essential oils [278,284,289,290,291,292,293,294,295,296,297,298,299,300,301,302], which have a number of remarkable biological effects, including analgesic, antibacterial, anti-inflammatory, antitumor, antioxidant and regenerating properties. As adjuvants, in our scientific group, compounds of the terpenoid, flavonoid and allylbenzene series (Figure 6—in the center from the bottom) have been extensively studied. The terpenoid adjuvants used by us demonstrated their ability to block intracellular efflux pumps that provide drug resistance to some bacteria due to the effective removal of antibiotics from the cell. Adjuvants also increase the permeability of bacterial cell membranes. The simultaneous delivery of an antibiotic and an adjuvant can increase the synergy of their action. For the combination of fluoroquinolone—a terpenoid—we observed a 2–3-fold increase in the effectiveness of the antibiotic (a 2–3-fold decrease in MIC) [280,283,285]. Recently, we showed that adjuvants (allylmethoxybenzenes, terpenoids, and flavonoids) have an enhancing effect on antibacterial drugs, including LF and MF, rifampicin, metronidazole, etc. [192,278,280,283,284,285].

However, the binding constants of both the antibiotics and adjuvants with the developed molecular containers—polymer ligands—are not high enough (about 10^4^ M), which will cause the dissociation of the complexes upon intravenous administration and will not provide a prolonged action of the antibiotic. So, we also created a moxifloxacin (fluoroquinolone) prodrug—a covalent conjugate of the antibiotic with mannosylated polymers (dendrimers) enhanced by a terpenoid adjuvant (limonene), with the function of prolonging drug action. Due to the application of such an “intelligent” prodrug system, selectivity was achieved: in microbiological experiments, an increased antibacterial effect on *E. coli* and *B. subtilis* cells was observed, while its effect on “good” *Lactobacillus cells* was reduced. We have developed pH, thermo- and stimulus-sensitive micelles [281,285,286], which are smart molecular containers that release drugs in a slightly acidic environment and in the presence of glutathione, corresponding to the microenvironment of tumors, or in an inflammatory focus, making them potentially applicable for antibacterial and anticancer drug delivery to macrophages. 

Our carbohydrate ligand, as well as the whole system in the complex—(ligand–stimulus-sensitive polymer–carrier–cyclodextrin) cargo from the drug and adjuvant, demonstrated the effectiveness both in cell cultures and in experimental models of infectious diseases in vivo. A study of antibacterial activity on bacterial cell cultures showed that the combined drug moxifloxacin or levofloxacin with an adjuvant is 3–5 times more effective than the original fluoroquinolone (a decrease in MIC due to the presence of an adjuvant and inclusion in a polymer carrier) and demonstrates a prolonged effect (retains activity for up to 8 days; the original fluoroquinolone loses effectiveness for 3 days). 

Study of pharmacokinetics in vivo has showed that mannosylated polymer systems based on mannan, cyclodextrins or polyethyleneimine increased the half-life of fluoroquinolones from the body of rats by more than 10 times and increased organ bioavailability, as well as the accumulation of the drug in the lungs by more than 7 times. 

Safety studies have demonstrated that mannosylated polymer systems are non-toxic with respect to cells of the line HEK293 (when using concentrations up to 100 micrograms per ml, they do not cause the thrombosis and hemolysis of erythrocytes).

Therefore, the creation of the pharmaceutical composition and structure of a targeted delivery system (a biocompatible polymer modified with cyclodextrin for drug loading and carrying an oligomannoside ligand of a complex structure) for the delivery of antibiotics to macrophages, due to its high-affinity interaction with mannose receptors (CD206), significantly increases organ bioavailability (bio-distribution and accumulation of drugs in the lungs) and enhances the permeability of drugs into bacterial cells. The use of adjuvants enhances the effectiveness of antibiotics by inhibiting efflux pumps (terpenoids and flavonoids). 

When the combined preparation of fluoroquinolone and its adjuvant are included in the developed delivery system, a dual mechanism of action of adjuvants is shown—an increase in the permeability of bacterial cells to the antibiotic and inhibition of the efflux proteins of bacteria (“throwing out” the drug from the cells)—which allows us to increase the accumulation of the drug in bacterial cells and reduce the load on healthy cells [251,252,253,254,255,256,257,258,259,260]. Potential perspectives of development include reducing the dosage of antibiotics, shortening the duration of treatment, and reducing the risk of developing resistance. 

The interaction of macrophages with ligands and the study of macrophage-derived particles have been investigated by several methods, including flow cytometry, fluorescence microscopy, and immunological methods. Additionally, we have developed some new approaches that allow us to study macrophages from a different point of view. We demonstrated the phenomenon of efflux in bacteria and eukaryotes and were able to inhibit it, which made it possible to increase the effectiveness of antibiotics or cytostatic drugs (Figure 6—bottom). The authors’ works present classical and original techniques (Table 5) of spectral approaches (FTIR, UV, NMR, fluorescence spectroscopy), computer modeling (molecular dynamics and neural network analysis), microscopy (atomic force, confocal scanning), biological and pharmacokinetic experiments to study both the fundamental aspects of biomimetics for pathogenic patterns recognized by macrophages, and practical applications to improve existing treatment regimens for macrophage-associated diseases. Using fluorescent methods, we studied the interaction of ligand receptors with living cells, adsorption and permeability over time, and the effect of efflux inhibitors on drug permeability and retention. FTIR spectroscopy was used for the high-throughput screening of lectin–ligand interactions using concanavalin A (Figure 6—top left) as a model mannose receptor to optimize the components and molecular architecture of a delivery system [192,276,278,279,283,303]. FTIR spectroscopy can potentially help to monitor the individual status of therapy: whether drug compositions affect the bacteria, macrophages, or tumor cells or whether they are indifferent. The technique allows testing new drugs or drugs in delivery systems. Recently, we have developed an original technique for detecting the selectivity of the action of drug formulations using FTIR. For example, using FTIR, we demonstrated the selectivity of chitosan-based micellar systems, and observed their effect on A549 cells and, conversely, on the protection of normal HEK293 cells [286]; a similar effect was observed for bacterial *E. coli* cells vs *Lactobacilli*. 

Macrophage-related biosensing system perspectives. FTIR spectroscopy provides valuable data on the interaction of cells with polymer systems, including the possibility to study the molecular mechanism of recognition, which opens prospects for the development of a biosensing system for detecting the activated pro-inflammatory macrophage (CD206+). On the other hand, with regard to biosensing using a macrophage membrane, we expect that FTIR spectroscopy will become a tool for studying the affinity of the receptor–ligand interaction. For biochip development, we used CD206+ macrophage membranes (membrane of macrophages dried on a polystyrene plate, then rehydrated and incubated with ligands). Macrophages are a difficult-to-grow cell culture that can be studied for several hours, so macrophage membranes as stable models are analytically significant. It turned out that CD206+ macrophage membranes demonstrate a binding ability similar to original cells. 

Thus, targeting macrophages using the biomimetics of pathogen patterns is a very effective strategy for creating therapeutic systems for a range of diseases. In addition, using macrophage-derived particles, it is possible to selectively target therapeutic cargo to tumor cells, which makes it possible to bypass biological barriers, “switch” the tumor microenvironment (hot/cold) and regulate the status of inflammation. In other words, targeting macrophages and using macrophage-derived membranes as drug carriers have huge prospects for creating a golden bullet for the treatment of infectious, oncological, neurodegenerative, and autoimmune diseases.

Despite the fact that macrophage-mediated systems can turn to be more preferable than traditional methods of drug delivery, their use in clinical practice may be limited by the effectiveness and possible side effects.

Clinical trials of the macrophages reprogramming into a cytotoxic phenotype and introduced to patients with cancer resulted in only a marginal therapeutic effect [312,313] and in some cases slight fevers and chills were observed as side effects [314]. Moreover, in accordance with the nature of macrophages, their administration may lead to immune responses and increase the level of proinflammatory cytokines [315], which together with the problem of the distribution of macrophages in healthy tissues [316] may increase the risk of several side effects.

The development of macrophage-derived particles for effective targeted treatment still remains a challenge, which is mainly due to their low stability or difficulties associated with the pharmacokinetics of therapeutic agents. At the moment, most strategies are based on the release of the drug through the penetration of the cell membrane [317] or exosomal release [318] once a macrophage-derived carrier reaches its target site. These methods of therapeutic cargo transfer still need to be developed, since an important task remains to maintain a balance between the timely release of the drug and its therapeutic activity. In this regard, bypassing the need for drug to leave the macrophage-derived carrier is a potent strategy, which is implemented in photothermal therapy [319].

Since macrophages are a part of immune system, drugs targeting macrophages may trigger multiple mechanisms, which lead to immune-related side effects [320]. At the moment, only few macrophage-targeting drug delivery systems have been approved for clinical use, which is due to the lack of knowledge of their biological properties [321], as well as due to their poor stability. Biomimetic systems based on macrophages may have great potential for scientific research and subsequent medical applications [322].

## 5. Discussion

The versatility, multifunctionality and plasticity of macrophages could make macrophage-mediated strategies a perspective tool for the treatment of inflammatory diseases, neurological diseases, infectious diseases and cancer. Macrophages’ inherent ability to penetrate a number of physiological barriers (Figure 1 and Figure 2), combined with possibilities to re-program their functionality, could yield new treatment modalities based on macrophages and their derivatives. The use of macrophage-derived particles is one of these options. Compared to erythrocyte-derived membranes, which have been studied for a similar purpose for several decades, they could provide much broader opportunities for drug discovery. The use of macrophage-derived particles allows us to employ the biological functions of macrophages (such as “Trojan horse”-approach). This could provide enhanced efficiency in the treatment of a wide range of macrophage-related diseases. 

Alternatively, a macrophage itself can serve as a target. Depending on the indication, their targeted re-polarization is a possible way to make a cold tumor hot, or, vice versa, to combat autoimmune diseases. Further research is required for the successful application of these developments in the clinic.

Once studied and understood in more detail, macrophage-mediated systems may show superiority over traditional delivery systems, such as liposomes, micelles, polymeric NPs, dendrimers, etc. (Table 3 and Table 4). 

One might imagine a diagnostic approach employing macrophages and their derivative gaining more traction. For example, various macrophage-associated features could be used to monitor the efficacy of treatment of certain diseases. In the future, using macrophages and ligands to them, biochips could be developed to diagnose and monitor the status of macrophage-associated diseases and study the effectiveness of therapy (which can be optimized for each patient—personalized). Macrophage-derived particles could help to defeat cancer by reprogramming tumors, and these particles are also promising for rheumatoid arthritis and neurodegenerative diseases.

## Figures and Tables

**Figure 1 biomimetics-08-00543-f001:**
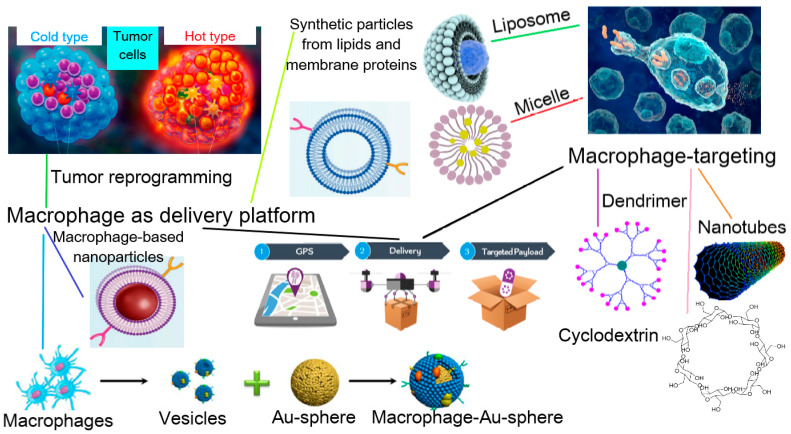
Macrophage-mediated strategies: use of macrophage-derived particles obtained ex vivo to deliver therapeutic agents; use of therapeutic agents designed for macrophage targeting in vivo.

**Figure 2 biomimetics-08-00543-f002:**
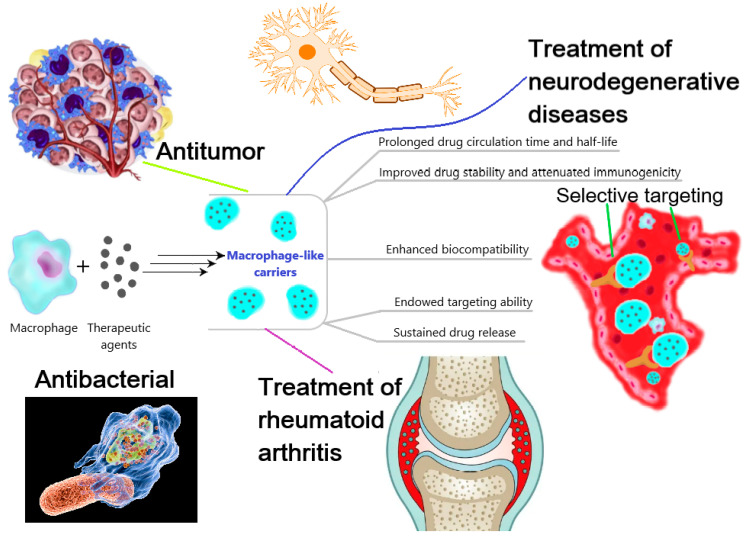
Advantages of using macrophage-derived-particle applications for drug delivery.

**Figure 3 biomimetics-08-00543-f003:**
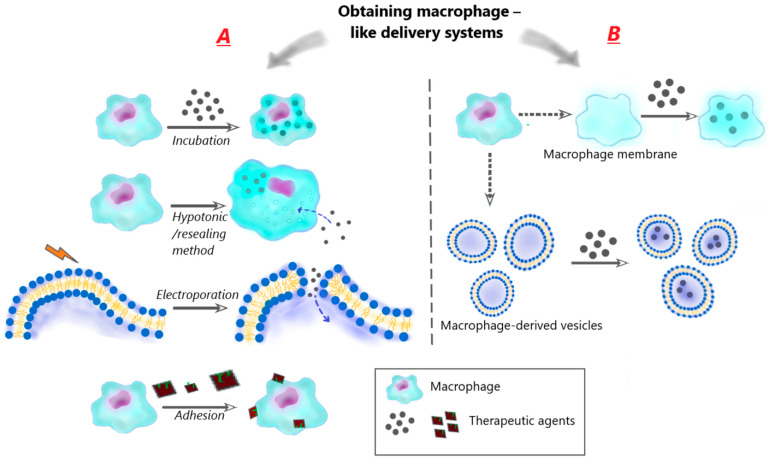
Two approaches to the use of macrophages in the development of delivery systems: (**A**) use of living cells (methods of incubation, hypotonic dialysis, electroporation and adhesion); (**B**) use of macrophage-derived membrane structures (cell membranes and vesicles).

**Figure 4 biomimetics-08-00543-f004:**
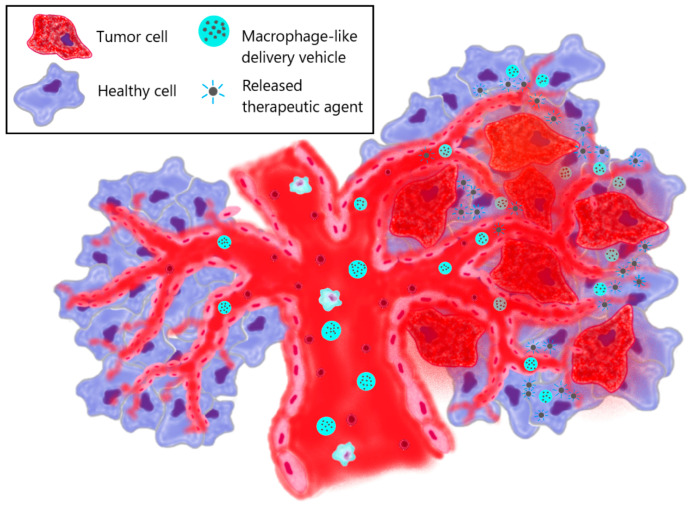
Illustration of the targeted ability of macrophage-derived carriers of anticancer therapeutic agents.

**Figure 5 biomimetics-08-00543-f005:**
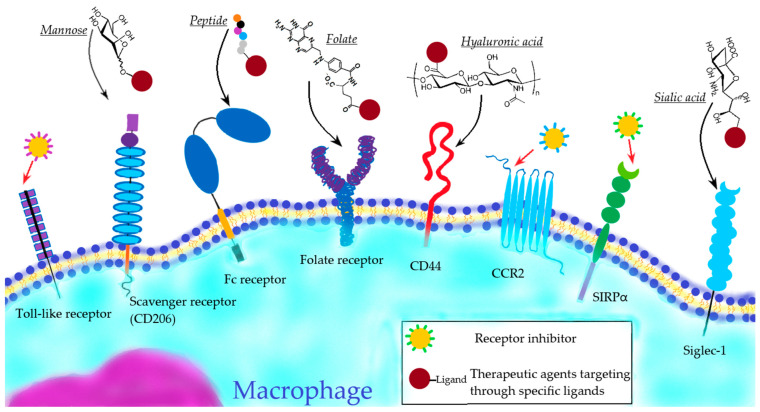
Active macrophage targeting via different macrophage receptors.

**Figure 6 biomimetics-08-00543-f006:**
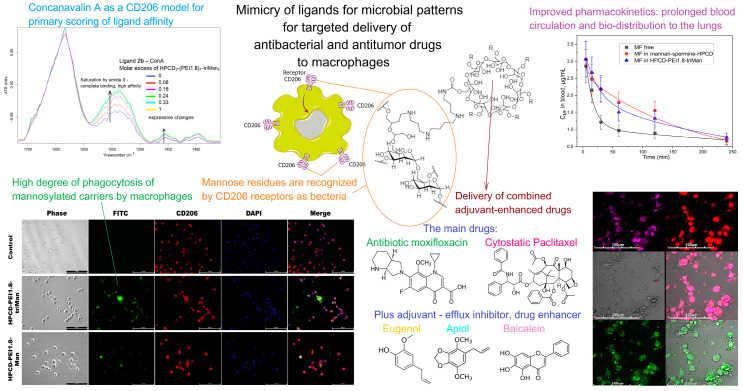
A brief presentation of the idea of macrophage targeting through their CD206 receptors by creating drug delivery systems that contain mannose residues mimicking pathogen patterns. The IR spectra of concanavalin A complexes (a model mannose receptor) with mannosylated polymer are shown at the **top left**—an example of primary screening for CD206 affinity. Confocal images of alone macrophages with FITC-labeled (green channel) mannosylated polymers are shown at the **bottom left**: we observed the greatest absorption by macrophages of particles with a trimannoside backbone, mimicking the oligosaccharides of bacteria. The **bottom center** shows the main drugs (antibacterial and anticancer drugs) that can be delivered to macrophages using our strategy, as well as their adjuvants (enhancers). Confocal images of macrophages with absorbed *E. coli* (as a model of intractable intramacrophage infection) are shown at the **bottom right**: pink—merged channels bacteria + doxorubicin. Due to the use of a high-affinity polymer to macrophages, the accumulation of the drug inside macrophages is increased by 4 times, and in addition, adjuvants (eugenol, apiol, etc.) inhibit efflux in bacteria and increase the penetration of the drug into bacteria. Polymer systems significantly increase the circulation time of moxifloxacin in the body of rats (**top right**) and increase the bio-distribution into the lungs to alveolar macrophages. Limitations of macrophage-mediated systems in terms of clinical translation barriers.

**Table 2 biomimetics-08-00543-t002:** Examples of current use of macrophages to design biomimetic drug-delivery system.

Utilization of Living Cells						
Method of Binding to Macrophages	Source	Carrier Particle	Cargo	Loading Efficiency (Related to the Drug)	Cell Viability	Refs.
Incubation (engulfment)	RAW264.7	-	Doxorubicin (400 µg/mL)	≈14% (after 10 s of incubation)	79% at 72 h after incubation	[83]
		Liposomes (diameter of 145 nm; composed of 1,2-dipalmitoyl-sn-glycero-3-phosphocholine and 1-myristoyl-2-stearoylsn-glycero-3-phosphocholine)	AuNRs (150 µg/mL) + Doxorubicin (25 µg/mL)	13.34% (after 6 h of incubation) 35.2%	85% after 6 h of incubation	[81]
		-	Bioengineered Salmonella typhimurium	220 ± 13 CFU/100 cells (after 60 min of incubation)	>90% after 60 min of incubation	[101]
	Mouse peritoneal macrophages	-	Doxorubicin (1–200 µg/mL)	No data	about 30–60% after 12 h of incubation	[102]
		Liposomes (diameter of 150 nm; composed of DPPC, DPPE, DPPG-Na and cholesterol)	Doxorubicin (1–200 µg/mL)	No data	about 80–90% at 12 h after incubation	
	BMM	Polymeric NPs (diameter of 0.35–2 µm; composed of PLGA) (100 µg/mL)	Nitric oxide	≈77% (after 2 h of incubation)	≈100% for incubation period of 24 h and 48 h	[90]
	Human monocyte-derived macrophages	Liposomes (size of 332 nm; composed of surfactants P188 and mPEG2000-DSPE) (100 μM)	Indinavir	85% (after 4 h of incubation)	No effect of drug encapsulation on macrophage viability was observed	[103]
Hypotonic dialysis	THP-1	-	Catalase (osmolality of 75.67 mOsm/L during 15 min of dialysis)	53%	89% after encapsulation	[72]
Electroporation	J774	-	Doxorubicin (20 mg/mL)	5% (after <20 s of electroporation)	Drug-loading significantly decreased cell viability	[73]
Adhesion	Raw 264.7	Multilayer microfilm (“backpack”) in disc-shaped polymer patches 7 µm in diameter (release region, magnetic region, payload region, and cell attachment region composed of BSM and JAC, PAH and MNP, PAH and CAT, and PAA and PAH-biotin, respectively)	Catalase (2.3 µU/cell backpack)	80% (after a brief incubation with the “backpacks”)	Attachment of cell backpacks to macrophages did not alter their major functions	[104]
	J774	Multilayer microfilm (“backpack”) in disc-shaped polymer patches 7 µm in diameter (composed of PMA, PNIPAAM, PAH, Chitosan and Hyaluronic Acid)	Bovine serum albumin	≈95% (after incubation with the “backpacks” for 4 h)	“Cellular backpacks” did not affect macrophage biological functions	[105]
**Application of Macrophage-Derived Membrane Structures**						
	**Source**	**Carrier Formulation**	**Cargo**	**Method of Encapsulation**	**Detected Proteins**	**Ref.**
Cellular membranes	J774	Polymeric NPs (diameter of 84.5 nm; composed of PLGA)	-	Sonication	CD126, CD130, CD120, CD119, CD14 and TLR4	[74]
	Mouse peritoneal macrophages	Polymeric macrophage-membrane-coated NPs (diameter of 82.3 ± 7.5 nm; composed of IGF1R-targeting polymer cskc-PPiP)	Paclitaxel	Sonication	No data	[106]
	RAW264.7	-	Methyltransferase like 14 + RS09	Coextrusion	No data	[107]
	RAW 264.7	Bi_2_Se_3_ hollow mesoporous NPs (diameter of 110 nm)	Quercetin	Coextrusion	α4 integrin, CCR2	[108]
Vesicles	RAW264.7	-	Paclitaxel	Sonication	Alix, TSG101, CD9, iNOS, Arg-1	[109]
	J774A.1	Liposomes (diameter of 100 nm; composed of L-a-phosphatidylcholine and Cholesterol)	Doxorubicin	Vortexing, sonication and coextrusion	CD81, CD63 and CD9	[99]
	RAW 264.7	Polymeric NPs (diameter of 96 ± 6.9 nm; composed of PLGA)	-	Sonication	CD45, CD14, CD44, CD18, Mac-1, etc.	[76]
	RAW264.7	-	Brain-derived neurotrophic factor	Simple mixing	Alix, Tsg 101, LAMP 2 and cytosolic protein β-actin	[110]

**Table 3 biomimetics-08-00543-t003:** Examples of the recent use of macrophages for anti-tumor therapy.

Vehicle	Carrier Formulation	Cargo	Target	Highlighted Features of Macrophage-Derived Particles	Therapeutic Effect	Refs.
Macrophage membrane	Chitosan NPs	-	Tumor cells: HeLa, MCF7 and MDA-MB-231 (in vitro)	Stability Biocompatibility and hemocompatibility Triggering apoptosis due to the presence of TNFα in macrophage membrane	Dose-dependent anti-tumor proliferative properties and triggering of apoptosis after 48 h of coculture	[100]
Macrophage	-	Doxorubicin	4T1 mouse breast cancer cells (in vivo)	Meaningful content of the drug High targeting ability	Significant inhibition of tumor growth and increasing the survival rate among tumor-bearing mice compared to saline and DOX groups after systemic injection for 15 days on days 6, 8, 10, 13 and 15	[83]
Macrophage	Poly(D,L-lactide-co-glycolide) micelles and pluronic block copolymer micelles	Paclitaxel	Human glioma cell line U87 (in vitro)	Main biological functions of macrophages were preserved Anti-tumor effect was enhanced compared to nano-paclitaxel	Significant tumor cell growth inhibition after 3 days of coculture	[130]
Macrophage	Poly(D,L-lactide-co-glycolide) NPs	Tirapazamine	4T1 mouse breast cancer cells (in vivo)	Targeting ability Enhanced accumulation in hypoxic areas of tumor	Inhibition of tumor growth and extension in the median survival time compared to saline and tirapazamine groups after two injections with an interval of 3 days. Especially high efficacy was attained in the synergetic chemotherapy.	[131]
Macrophage	-	siRNA lipoplexes	MDA-MB-468 breast cancer model (in vivo)	Ability for horizontal gene transfer of siRNA in tumor site Anti-tumor effect was enhanced compared to pure siRNA Results indicated that exosomal secretion via M2 activation is involved with gene transfer	A significant reduction in the tumor spheres growth after single administration (no control group)	[132]
Macrophage	N-methacryloyl mannosamine (conjugated to macrophage surface)	Nucleic acid aptamers	CCRF-CEM tumor cells (in vitro)	Surface modification did not affect macrophage phenotype and viability The capture of tumor cells was improved	High anticancer immune response via macrophages was observed after 30 h of coculture	[133]
Macrophage	-	Oncolytic adenovirus	Human prostate tumor model (in vivo)	Targeting ability Accumulation in hypoxic/perinecrotic areas of the tumor	A lasting antitumor effect, enhanced in comparison with saline group, with negligible metastatic frequency was observed after 48 h of single injection	[134]
Macrophage membrane	Gold nanoshells (AuNSs)	Cy7	4T1 cancer cells (in vivo)	Active targeting ability High tumoritropic accumulation Good biocompatibility Prolonged circulation time Membrane coating did not affect NIR optical properties of AuNSs	Effective inhibition of tumor growth and its complete eradication after systemic daily injection with NIR irradiation for 25 days. Antitumor effect was enhanced in comparison with Cy7-AuNSs and saline groups	[135]
Macrophage	Liposomes	AuNRs + Doxorubicin	4T1 mouse breast cancer cells (in vivo)	High targeting ability Effective infiltration into the tumor tissue High thermal sensitivity Controlled drug release via photothermal performance	Synergetic chemo- and phototherapy allowed enhanced tumor growth inhibition compared to pure liposomes and saline groups after 24 h of single injection	[81]
Macrophage	Liposomes	ICG (photothermal agent) + Resveratrol (anti-inflammatory drug)	4T1 post-operative model (in vivo)	Tumor-targeting ability Good inflammatory tropism Release of the liposomes was enhanced due to membrane destruction via phototherapy Excellent photothermal performance	Ablation of residual tumor tissues, inhibiting tumor postoperative relapse and reduction in postoperative inflammation. The inhibition of tumor growth was enhanced with the delivery of macrophage-derived particles compared to liposome and saline groups after systemic injection, following NIR irradiation every 2 days for 29 days	[136]

**Table 4 biomimetics-08-00543-t004:** Examples of utilizing macrophage-targeting therapeutic agents.

Receptor Targeting	Carrier Formulation	Ligand Modification/Coating	Cargo	Purpose	Result	Refs.
Mannose receptor	Liposomes	Mannose	DNA	Stimulation of immune response	Mannosylated cationic liposomes exhibited significantly improved DNA delivery compared to unmodified liposomes	[167]
	Polymeric micelles		siRNA	TAM repolarization	Modified micelles could selectively deliver efficacious amounts of functional siRNA into TAMs	[168]
	Liposomes		^64^Cu	PET imaging of TAMs	Highly selective accumulation of the liposomes in TAMs was observed	[169]
	Selenium NPs		Isoniazid	Treatment of tuberculosis	The NPs preferentially entered macrophages and accumulated in lysosomes, releasing isoniazid	[170]
Galactose receptor	Dextran NPs	Galactose	CpG, anti-IL-10 and anti-IL-10 receptor oligonucleotides	TAM repolarization	NPs accumulated in the tumor and was taken up predominantly by TAMs	[171]
	Chitosan-cysteine NPs		siRNA	Treatment of ulcerative colitis	Galactose modification significantly facilitated the uptake by macrophages and targeting ability of the NPs	[172]
	Poly(lactic-co-glycolic acid) NPs		Dexamethasone	Development of the strategy to catch macrophages during intestinal inflammation	NPs were effectively captured by macrophages	[173]
Dectin-1	Polymer–lipid hybrid NPs	Yeast cell wall microparticles, containing β-1,3-D-glucan	Cabazitaxel	Development of oral targeted drug delivery	The microparticles were rapidly and efficiently taken up by macrophages	[174]
	Mesoporous silica NPs		Doxorubicin	Development of anti-tumor therapy	Drug delivery to macrophages was enhanced compared to uncoated silica NPs	[175]
Fc receptor	Alginate NPs	Tuftsin	DNA	Development of anti-inflammatory agents	Tuftsin-modified NPs were rapidly internalized in murine macrophages	[176]
Folate receptor-β (FRβ)	-	Anti-mouse FRβ monoclonal antibody	Pseudomonas exotoxin A	TAM depletion	Direct eliminating of TAMs was attained	[177]
	Poly(amidoamine) dendrimers	Folic acid	Methotrexate	Alleviating of the inflammatory disease of arthritis	High degree of specific binding and internalization of the dendrimers into macrophages was observed	[178]
	Human serum albumin nanocapsules		-	Evaluating targeting ability of folic acid-modified agents	The internalization of nanocapsules was enhanced via FR specificity	[179]
CD44	Hyaluronic acid–tocopherol succinate micelles	Hyaluronic acid	Rifampicin	Development of tuberculosis treatment	Micelles exhibited significant phagocytosis and a CD44-dependent uptake in comparison to free drug	[180]
	Liposomes		Prednisolone	Development of rheumatoid arthritis therapy	Enhanced cellular uptake, mainly mediated by caveolae- and clathrin-dependent endocytosis, was achieved	[181]
	Poly(lactic-co-glycolic acid) NPs		Curcumin	Alleviation of ulcerative colitis	Enhanced drug delivery to intestinal macrophages and selective accumulation in inflamed colitis tissue with minimal accumulation in healthy colon tissue was observed	[182]
Siglec-1	Liposomes	Sialic acid	Epirubicin	Tumor therapy	The tumor-targeting efficiency and the accumulation of epirubicin in monocytes was improved compared to unmodified liposomes	[183]
			Zoledronic acid	TAM depletion and repolarization	High targeting ability was observed	[184]

**Table 5 biomimetics-08-00543-t005:** The approaches for modeling polymer drug delivery systems’ interactions with macrophage-like systems and studying and their effects on cells.

Method	Applications	Brief Description	Refs.
FTIR spectroscopy	Macrophage CD206 receptor—ligand interaction studies on the example of the ConA model and mannosylated polymers	The use of a model receptor protein allows for the rapid primary screening of ligands and selection of the most affine ones, and it is not necessary to isolate hard-to-reach CD206	[278,283]
	Drug—delivery system (to macrophages) interactions	Registration of FTIR spectra of drug complexes with different polymer ratios and calculation of dissociation constants and entrapment efficiency. Study of molecular details of binding (functional groups)	[283,284,286]
	Cell—drug formulation interactions. The effect of the drug on the cells. Selection of the optimal composition of the drug formulation	Provide information about the main components of the cell interacting with the drug. Using this technique, efflux and its inhibition on bacterial and cancer cells were demonstrated	[277,286]
	Quantification of living cells	Centrifugation of cell suspension and registration of the FTIR spectra of sediment. Low analysis time: does not require seeding of bacteria on a Petri dish	[280]
	Characterization of polymeric drug delivery systems	The presence of all components, and the success of crosslinking. Molecular architecture	[192,278,280,281,282,283,284,285,286,287]
NMR spectroscopy	Drug interaction with the delivery system	The NMR spectrum provides valuable information about the functional groups involved	[284]
	Characterization of polymer drug delivery systems	The presence of all components, and the success of crosslinking	[192,280,283]
Fluorescence spectroscopy	Macrophage CD206 receptor—ligand interaction studies on the example of ConA model and mannosylated polymers	Quenching of tryptophan fluorescence in the receptor protein and an increase in fluorescence anisotropy during ligand binding. An alternative is using a FITC-labeled ligand	[279]
	Inclusion of fluorophore drugs in polymer particles	Change in fluorescent properties, the position and intensity of the maximum, as well as FRET	[304]
	Interaction of ligands with cells, adsorption and permeability over time, and the effect of efflux inhibitors on drug permeability and retention		
UV spectroscopy	Macrophage CD206 receptor—ligand interaction studies on the example of ConA model and mannosylated polymers	Change in protein uptake during ligand binding and change in secondary structure	[279]
	Loading and release of drugs from polymer carriers	Absorption characterizes the amount of drug loaded or released from nanoparticles	[278,285]
	Antibacterial activity	A600 correlates with the number of colony-forming units	[280,284,285]
Circular dichroism spectroscopy	Secondary structure of macrophage CD206 receptor (or its model protein on the example of ConA model) during ligand binding	Changing the circular dichroism sometimes with a cardinal reversal of the spectrum	[305]
	Loading of chiral drugs into polymeric particles		[306]
Isothermal titration calorimetry	Study of macrophage CD206 receptor—ligand interaction studies on the example of ConA model and mannosylated polymers	Thermodynamic parameters (enthalpy, entropy and Gibbs energy) of ligand–receptor complex formation	[307,308,309,310]
Atomic force microscopy, SEM and TEM	Study of the morphology of nanoparticles, simulating epitopes of pathogenic microorganisms recognized by macrophages. Study of the morphology of bacterial and macrophage cells with adsorbed polymers	High-quality images providing information about the structure of nanoparticles and their effect on bacteria	[281,304,306]
Nanoparticle tracking analysis (NTA)	Characterization of macrophage target drug delivery system (nanoparticles)	The rate of particle movement is related to a sphere equivalent hydrodynamic radius as calculated through the Stokes–Einstein equation	[285]
Dynamic light scattering	Detection of polymeric nanoparticles interaction with cells surface by changing of zeta potential of bacteria and macrophages cells during polymer adsorption	The zeta potential characterizes the stability of nanoparticles. For cells, there is a recharge during the adsorption of polymers	[192,281,304]
Confocal laser scanning microscopy	Interaction of drug formulations with bacterial and eukaryotic (macrophage and cancerous) cells	Images from multiple cells at the micro and nanoscale. Inhibition of efflux (reverse release of drugs from cells) has been demonstrated	[192,277,311]
Microbiological studies	Study of the antibacterial effect of drugs, including effect on bacteria inside macrophage	The strengthening and prolonged (in vitro) of antibacterial drugs due to the addition of adjuvants to them has been demonstrated	[192,278,280,283,284,285]
Pharmacokinetics studies	Testing the macrophage-targeted drug delivery system in terms of the drug circulation time in the bloodstream and bio-distribution	A multiple increase in the half-life of the drug is shown, especially for covalent pro-drugs, and accumulation in the lungs	[280,283,285]
Flow cytometry	The existence of fluorescent nanoparticles with the drug (not debris)		[281,304,306]
	Nanoparticles adsorption on *E. coli* cells, and quantification of living cells by DAPI staining		
Computer modeling	Molecular dynamics and neural network analysis of macrophage CD206 ligand and drug–polymer interaction	The study of ligands does not require synthesis in the laboratory and complex experiments—as does the primary stage of selecting candidates for drug delivery systems to macrophages. Molecular architecture of complexes, binding sites and prediction of binding energy.	[276,288]

## Data Availability

The data presented in this study are available in the main text.

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
