# Peer review of "Biomimetic Systems Involving Macrophages and Their Potential for Targeted Drug Delivery"

_biomimetics, 2023, doi:10.3390/biomimetics8070543_

Round 1

Reviewer 1 Report

Comments and Suggestions for Authors

      In this research, the authors reviewed the status of biomimetic systems involving macrophages and their potential for targeted drug delivery. In my opinion, the current version of this manuscript fits the scope of Biomimetics and could be accepted after major revision.

My specific comments are in detail listed below:

1.     The quality of Figure 1 and Figure 2 was not that clear. A clear version may be better.

2.     In this review, the advantage of using biomimetic systems involving macrophages should more clearly compared with the normal liposome or albumin nanoparticles.  Some references should be added to this part including 10.1002/adma.202206121.

3.     Some minor mistakes exist in the references. The authors should correct it.

4.     In this review, the merits of currently used tumor biomimetic systems involving macrophages for immune regulation should be pointed out, like PD-L1, VEGF, and et al. Some references should be added to this part including 10.1016/j.ijbiomac.2022.10.167.

5.     The clinical transformation barriers of biomimetic systems involving macrophages should be better out-looked.

Reviewer 2 Report

Comments and Suggestions for Authors

In this review paper, the authors presented "Biomimetic systems involving macrophages and their potential for targeted drug delivery". From my point of view, the topic is fascinating. The manuscript is very informative, concise, and well-written. Therefore, I recommend its publication with minor revision.

Following are my suggestions:

1)      The authors should discuss the drawbacks of using macrophages as biomimetic systems.

2)      The authors should also discuss the future prospects of this work.

Reviewer 3 Report

Comments and Suggestions for Authors

In this review, Savchenko and colleagues provide a comprehensive overview of two biomimetic strategies related to macrophages for therapeutic applications. First, the authors described utilization of certain types of macrophages, including tumor-associated ones, or macrophage-like particles for targeted drug delivery. Second, the authors discussed about therapeutic agents that naturally bind to macrophage-specific ligands can be employed. Overall, the review would benefit from a more critical discussion of the potential drawbacks and solutions to such drawbacks for the clinical translation of macrophages -mediated platform. This reviewer recommends acceptance of this manuscript upon consideration of the following points:

Additional Comments:

1.     The authors could consider adding a separate section for the discussion and summary or future perspective. 

2.     All abbreviations should be fully spelled out on first appearance. For instance, TAMs is first mentioned in Table 3; page 17 but the full name appears in page 24. The author should check all abbreviations throughout the manuscript.

Comments on the Quality of English Language

Many grammatical errors throughout the manuscript need to be corrected. There are plagiarism detected, which must be corrected.

Reviewer 4 Report

Comments and Suggestions for Authors

In the review, authors provided detailed summarization of the application of macrophage related systems in drug delivery. The topic is interesting, and covers a broad field. Therefore, it can be accepted after proper revision.

1.     Authors should provide a section named “conclusion and perspective” to discuss the future of this field.

2.     In section 3.3, the membrane from different phenotype of macrophage has different targeting capacity. Recent studies have showed the membrane from M1 macrophage has better tumor targeting capacity (Biomaterials, 2020, 255, 120159; J Control Release, 2020, 321, 589-601), authors should refer them.

3.     The function of macrophage in immunosuppressive microenvironment should be discussed in section 2, and related review is suggested to refer (Acta Pharm Sin B, 2020, 10(11), 2156-2170).

4.     The function of macrophages in chronic diseases should be discussed in section 2, and related review is suggested to refer (Chinese chemical letters, 2022, 33 (2) , pp.597-612).

Reviewer 5 Report

Comments and Suggestions for Authors

The review article by Savchenko et al gives a wholistic overview of the biomimetic strategies for targeted biomolecule delivery using macrophages. The review is timely and well written. The concepts described here can benefit a larger audience who are working towards macrophage-based immunomodulation.

Here are few suggestions which the authors should consider for improving the manuscript:

1.     Section 2.1, Macrophage phenotypes such as M2a, M2b, M2c need to be explained. Cell surface markers, cytokine/ remodelling enzymes secreted by them should be outlined for a better understanding. Their roles in foreign body response or tumour or disease related environment needs to be discussed further.

2.     Section 3.1, describes various sources of macrophages which have been reported as delivery systems. A critical note on how this can be translated need to be added. For instance, the yield of peripheral blood monocytes (PBMCs) and their polarization for such application need to be discussed.

3.     In Section 3.1, which describes various strategies for obtaining macrophage-like particles, a critical note on how to make these particles regulatory compliant conducive for clinical translation need to be added. For instance, the cytokines/ drugs or culture media conditions can be Xenogenic, how will these factors affect the translational prospects?

4.     A note on the stability of such macrophage-like particles can be added.

5.     Section-4 describes macrophage-targeting strategies, however, the fate of engineered macrophages or macrophage-like particles after targeting need to be discussed. Whether they get recycled? What would be the cell-mediated immune response after the targeting functions is perfomed?

6.     The review can benefit from a ‘Future perspective/ directions’ section. Can biomaterials-based delivery strategies be used/ coupled with such engineered macrophages? Can other sources be explored other than patient specific-PBMCs? May be iPSCs? How’s CAR T strategy different from engineered macrophages? This could add an interesting direction to the readers.

Comments on the Quality of English Language

The article is written satisfactorily without any concerns on the quality of English language.

Reviewer 6 Report

Comments and Suggestions for Authors

The authors proposed a review of the macrophage contribution to drug delivery strategies. Although this is a hot-topic theme, the manuscript is poorly organized providing very general and redundant content. The authors enumerate molecules, diseases, etc. but do not explain the mechanism or provide context or goal, which results in a very general overview. Figures do not add detail, inspire new ideas, or contribute to increasing the knowledge of the theme. Also, some information provided is not accurate. Thus, the manuscript requires major revisions and it is not suitable for publication in its present form.

1.      Please revise the abstract for scientific accuracy and correctness. For instance, words such as “biological objects” do not seem appropriate, and “macrophage-like particles” do not constitute a type of macrophage.

2.      The Sections 1 to 3 should be revised and merged. It is not clear if the focus of the manuscript is on infectious, inflammatory, or oncology diseases. A brief description of the involvement of macrophages and the mechanisms of action in these diseases should be provided. References are missing to support the statements.

3.      Why is the phagocytic function of macrophages emphasized in almost all sections of the manuscript, sometimes more than once in the same paragraph? E.g. section 2.2: “M1 macrophages are the most characterized subpopulation and are known primarily  for their phagocytic function [20,21]. (…) The main functions of M1 macrophages are phagocytosis, utilization of the remnants of destroyed cells (…)”.

4.      Please clarify the meaning of: “utilization of pathogens and necrosis products of cells”. Utilization for what purpose?

5.      Figure 3 – Please indicate the advantages/ disadvantages of each method, and for what purpose (disease, therapeutic agent, …) they are being investigated.

6.      Table 1. – What was the rationale for the selection of the examples? Properties of the drugs? Application? Please complete the table with the dimensions and composition of the drug carrier. Please clarify if the loading efficiency presented is related to the carrier or to the cell uptake. Also, the efficacy of the cargo is as important as the loading efficiency thus cell responses should be indicated.

7.      Figures with data from multiple publications to support the written information are missing from the review manuscript and should be included. In general, the illustrations provided are too simple for a scientific review.

8.      Table 2. Please indicate if the drug was single or multi-dose administered and the outcomes should be compared with the ones from a control condition.

9.      I suggest the authors reorganize the manuscript to avoid repetition in sections. For instance, the fact that tumors are first referred to in some sections and in others appear last, makes the reading difficult to follow.

10.   FTIR, NMR spectroscopy, etc. are techniques not approaches for modeling drug systems. This is not accurate and it is not scientifically correct.

11.   Please include a paragraph on the final remarks/conclusions of the manuscript.

Comments on the Quality of English Language

Minor editing of the English language is required.

Round 2

Reviewer 3 Report

Comments and Suggestions for Authors

No further change needed.